# Y–SHAPED GENERATIVE FLOWS

## ABSTRACT

Modern continuous-time generative models often induce *V-shaped* transport: each sample travels independently along nearly straight trajectories from prior to data, overlooking shared structure. We introduce *Y-shaped generative flows*, which move probability mass together along shared pathways before branching to target-specific endpoints. Our formulation is based on a novel velocity-driven objective with a sublinear exponent (between zero and one); this concave dependence rewards joint, fast mass movement. Practically, we instantiate the idea in a scalable neural ODE training objective. On synthetic, image, and biology datasets, Y-flows recover hierarchy-aware structure, improve distributional metrics over strong flow-based baselines, and reach targets with fewer integration steps.

## 1 INTRODUCTION

Recent advances in generative modeling are centered on the paradigm of continuous-time flows, where a velocity (vector) field is learned to transport mass from a simple prior distribution to a complex target distribution (Ho et al., 2020; Lipman et al., 2022). State-of-the-art approaches such as flow matching explicitly optimize straight-line trajectories, treating each data point as moving independently along its own path (Tong et al., 2023; Liu et al., 2022). For instance, transporting a source point $x_0$ to two

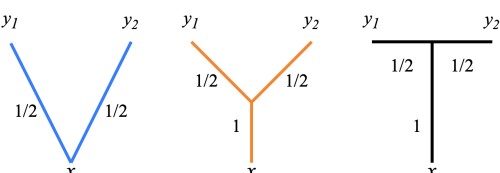

Figure 1: *Blueprint of flow shapes*. Conceptually in a V-flow, the mass separates and moves source from $x$ along straight lines. In Y and T flows, the mass moves together and then splits into targets.

distinct targets $y_1$ and $y_2$ is realized through two separate, non-interacting trajectories. This pattern corresponds to a V-shaped flow, since the objective provides no mechanism for trajectories to coordinate, merge, or share transport cost. One of the central problems that current V-shaped continuous-time approaches aim to solve is to reduce the number of steps to reach the target. To generate data, we must simulate an Ordinary Differential Equation (ODE), which is computationally expensive and often requires many steps. To reduce the number of ODE steps, several techniques have been introduced, such as mean flows (Geng et al., 2025), shortcut models (Frans et al., 2024), Schrödinger bridges (Gushchin et al., 2024), and *jumps* (Holderrieth et al., 2024).

But is the translation between simple and complex data distributions well-described by a uniform V-shaped flow? Can the V-shaped flow pattern be efficiently *shortcutted*? We argue that this is not always the case, because these approaches overlook the fact that real-world data possesses hierarchical and taxonomic structures. A more natural formulation is to view generation through flow models that adapt their number of steps depending on the samples we aim to generate. Samples in central modes of the target distribution can be reached in a few steps, whereas more diverse or peripheral samples may require additional ones. This flow would follow a common path for different points up to a mode or certain moment and then split toward the target. We propose to study different shapes of generative models that can capture the notion of mapping from the general to the specific, see Figure 1.

Building such generative models remains an open challenge that we aim to address. To this end, we draw inspiration from branched transportation theory, which is motivated by a wide range of natural and engineered structures such as vascular systems, trees, river basins, and urban planning (Buttazzo & Stepanov, 2003; Xia, 2003; Bernot et al., 2005). These structures arise from an economy of scale: transporting a large mass together is proportionally cheaper than splitting it into many smaller units (Santambrogio, 2015, Section 4.4). Consequently, efficient flows naturally exhibit branching geometry. Translating this principle into generative modeling suggests that trajectories should allow

data points to merge toward well-represented modes and subsequently branch out to reach diverse targets. In contrast to conventional V-flows, we propose *Y-shaped generative flows*, where mass can aggregate and then adaptively separate, capturing the structure of the data.

**Contribution.** We introduce *Y-shaped generative flows*, continuous-time generative models that favor shared transport before branching grounded in a *velocity–power* action which is concave in the speed and thus promotes concentration and subsequent branching of trajectories, while avoiding fractional powers of the density. Building on this, we develop a practical neural–ODE training objective that trades off the time-integrated branched action with a *Sinkhorn* boundary divergence to enforce endpoint matching, yielding a mesh-free and scalable procedure. Empirically, on synthetic, image, and single-cell datasets, our models learn branched trajectories and improve metrics (e.g., $W_1$/$W_2$/MMD) compared to strong flow-based baselines.

## 2 BACKGROUND

**Notation.** Let $\Omega \subset \mathbb{R}^d$ be a compact and convex domain. We denote by $\mathcal{P}_2(\Omega)$ the space of probability measures on $\Omega$ with a finite second moment. When a measure $\mu$ admits a density relative to the Lebesgue measure, we denote its density by $\rho$ (i.e., $d\mu(x) = \rho(x)dx$). Vectors are column vectors, $|\cdot|$ is the standard Euclidean norm, and $\nabla\cdot$ is the divergence operator and $|\cdot|_F$ is the Frobenius norm. The notation $d\mathcal{H}^1$ represents the differential element with respect to the *1-dimensional Hausdorff measure*. In practical terms, for a curve or any 1-dimensional set in space, the measure $\mathcal{H}^1$ simply calculates its length.

**Continuous Normalizing Flows (CNFs)** proposed by Chen et al. (2018) are generative models that define a probability density path through a neural ordinary differential equation (ODE) to morph one probability distribution, $\mu_0$, into another, $\mu_1$: $\frac{d}{dt}x_t = v_\theta(x_t, t)$, where $x_{t=0} \sim \mu_0$. Let $\Phi_t$ be the flow map associated with this ODE, which transports a particle from its initial condition at time $0$ to its location at time $t$. The pushforward density $\rho_t = (\Phi_t)_\# \rho_0$ evolving under this dynamics *necessarily* satisfies *continuity equation* with the parameterized velocity field $v_\theta$:

$$\partial_t \rho_t + \nabla \cdot (\rho_t v_t) = 0 \quad \text{on } \Omega \times (0,1), \tag{1}$$
$$\rho_{t=0} = \rho_0, \quad \rho_{t=1} = \rho_1.$$

A key result is the instantaneous change of variables formula, which describes how the log density evolves along a trajectory: $\frac{d}{dt} \log \rho_t(x_t) = -\nabla \cdot v_\theta(x_t, t)$. This allows for a likelihood calculation by integrating this quantity over time. Training can be done by directly maximizing likelihood.

**Monge–Kantorovich Optimal Transport (OT)** seeks a way to morph one probability distribution with minimal effort, as quantified by a cost function $c(x, y)$. The Monge formulation seeks a deterministic map $T : \Omega \to \Omega$ that pushes $\mu_0$ to $\mu_1$ and minimizes the total cost $\int c(x, T(x))d\mu_0(x)$ (Villani, 2008). This problem can be ill-posed, its relaxation, the Kantorovich problem, searches over *couplings* (joint distributions) $\pi \in \Pi(\mu_0, \mu_1)$ with marginals $\mu_0$ and $\mu_1$: $\inf_{\pi \in \Pi(\mu_0, \mu_1)} \int_{\Omega \times \Omega} c(x, y)\, d\pi(x, y)$. For cost $c(x, y) = \|x - y\|^2$, the square root of the solution is the Wasserstein-2 distance.

**Benamou–Brenier OT** frames transportation as a continuous-time problem. For flow $\Phi_t$, we have a pushforward density $\rho_t = (\Phi_t)_\# \rho_0$ evolving under the velocity field $v$. The Benamou–Brenier theorem states that the squared Wasserstein distance equals the minimal *kinetic energy* of such a flow:

$$W_2^2(\mu_0, \mu_1) = \inf_{\rho_t, v_t} \int_0^1 \int_\Omega |v_t(x)|^2 \rho_t(x)\, dx\, dt \quad \text{s.t. Eq. 1}. \tag{2}$$

The minimizers of this problem are constant-speed geodesics in the Wasserstein space. When an optimal transport map $T$ exists for the static problem (e.g., when $\mu_0$ is absolutely continuous), the geodesic is given by *interpolation*: $\rho_t = ((1-t)\mathrm{Id} + tT)_\# \rho_0$. The corresponding velocity field is constant along the trajectories, $v_t(x_t) = T(x_0) - x_0$.

## 3 FLOW SHAPE CONTROL

*What is the most natural way to control the shape of a generative flow?* To approach this question, we draw inspiration from the mathematical domain that studies the organization of branching structures in nature and human-made systems, such as rivers, blood vessels and traffic networks. Branched

optimal transport serves as an umbrella term for a family of methods designed to capture such irrigation phenomena (Xia, 2003; Brasco et al., 2011). The core of all branched methods is a concave transport cost. Typically of the form $m^\alpha \ell$, where $m$ is the mass and $\ell$ is the distance, and $\alpha \in (0, 1)$ is a branching coefficient. Since $(m_1 + m_2)^\alpha < m_1^\alpha + m_2^\alpha$ for $\alpha < 1$, this cost structure incentivizes the masses to travel together along shared paths. Transporting a combined mass along a single path is cheaper than transporting a mass $m$ along a few separate parallel paths (Santambrogio, 2015).

In one of the main branched transport formulations introduced by Xia (2003), the transport path is represented by a vector measure $F$, which can be understood as a flow network. Physically, $F$ corresponds to the momentum of the flow at each point, encoding both the direction and the amount of mass being moved. Transport occurs along a network of one-dimensional paths, denoted by the set $M$, where $\theta$ represents the volume of mass flowing along each path. The evolution of mass is governed by the continuity constraint $\nabla \cdot F = \mu_0 - \mu_1$. The divergence, $\nabla \cdot F$, measures the net outflow from any point. Finally, $M^\alpha(F)$ serves as the cost function that we seek to minimize:

$$M^\alpha(F) = \int_M \theta^\alpha \, d\mathcal{H}^1.$$

So, this integral calculates the total mass moved multiplied by the distance it travels. However, this cost is highly non-convex and defined over a complex space of singular, 1-dimensional vector measures $F$. Directly minimizing this functional is computationally intractable.

Another formulation inspired by the Modica–Mortola (MM) framework was proposed by (Oudet & Santambrogio, 2011). It replaces the singular, non-smooth branched transport problem with a sequence of regularized, elliptic energy functionals $M_\lambda^\alpha$ defined over the more regular space of $\mathcal{H}^1$ vector fields. For a given $F(x)$, the approximating functional is:

$$M_\lambda^\alpha(F) = \lambda^{\gamma_1} \int_\Omega |F(x)|^\alpha dx + \lambda^{\gamma_2} \int_\Omega |\nabla F(x)|^2 dx, \tag{3}$$

where $\lambda > 0$ is a small parameter. The exponents $\alpha$, $\gamma_1$, and $\gamma_2$ are derived directly from the transport dimension $d$ and the cost exponent $\alpha$. But this method is practically unstable and difficult to scale for real-world generative modeling applications. Please, see Appendix ??.

Another perspective on branched optimal transport is the Benamou–Brenier formulation studied by Brasco et al. (2011), which is also equivalent to the formulation of Xia (2003). The idea is to consider the Benamou–Brenier optimal transport problem 2, but with a *concave* cost function. With $\alpha \in (0, 1)$ being the branching parameter, we define the cost that measures the *work* of moving a mass $m$ over a distance $\ell$, simply written as $m^\alpha \ell$. If $\alpha = 1$, the cost becomes linear in the mass ($m\ell$), recovering the classical OT setting 2. Identifying the length $\ell$ with the velocity $v$ and the mass $m$ with the time-dependent probability density $\rho_t$, the dynamic branched transport is:

$$B^\alpha(\mu_0, \mu_1) = \inf_{\rho, v} \int_0^1 \sum_{i \in I_t} |v_{t,i}| \, (\rho_{t,i})^\alpha \, dt, \quad \text{s.t. Eq. 1}. \tag{4}$$

In this formulation the cost 4 need to be finite, the measure $\rho(t, \cdot)$ must be *purely atomic* for (almost every) time $t$. This means that the mass is concentrated at a countable set of points $\{x_{t,i}\}_{i \in I_t}$: $\rho(t, \cdot) = \sum_{i \in I_t} \rho_{t,i} \, \delta_{x_{t,i}}$, where $\rho_{t,i} = \rho(t, \{x_{t,i}\})$. The associated momentum is then $F = \rho v = \sum_{i \in I_t} v_{t,i} \, \rho_{t,i} \, \delta_{x_{t,i}}$, where $v_{t,i} = v(t, x_{t,i})$ is the velocity of the atom $x_{t,i}$.

Both the original formulation of Xia (2003) and the branched Benamou–Brenier formulation (Brasco et al., 2011) are computationally impractical for high-dimensional problems. Incorporating a mass-dependent term in the cost function, especially with a sublinear exponent $\alpha < 1$, significantly increases computational complexity and limits scalability. This motivates our development of a more efficient relaxation.

> ***Takeaway:*** *To control the shape of flows, branched optimal transport use $m^\alpha \ell$, where $m$ is the mass and $\ell$ is the distance, and $\alpha \in (0, 1)$ is a branching coefficient. Since $(m_1 + m_2)^\alpha < m_1^\alpha + m_2^\alpha$ for $\alpha < 1$, this cost structure incentivizes the masses to travel together along shared paths (Santambrogio, 2015). This is very useful idea, but running these methods in practice is often not feasible in the continuous case.*

## 4  METHOD

To address the challenge of complex density optimization in branched generative models, we present a simple alternative: placing the nonlinearity on the *velocity* rather than on the *mass*. We begin by examining the properties of the flux-based branched formulation. Let us first decompose the components of the Modica-Mortola approximation referenced in Eq.11. The components are:

**Concave Flux Term** ($\int |F|^\alpha$). This term forces the transport density to concentrate on a lower-dimensional set, promoting sparsity rather than diffusing throughout the space. Minimizing a concave power encourages $F$ to be 0 (empty space) or concentrated on a lower-dimensional manifold, effectively forming the transport network.

**Dirichlet Regularizer** ($\int |\nabla F|^2$). This term penalizes sharp transitions, preventing paths from collapsing into infinitely thin Dirac masses. It forces the flux to spread over a strip of finite width $A$, creating a "corridor." Without this regularization, the concave flux term would prefer infinitely thin, infinitely dense paths, which are ill-defined in the functional space.

To obtain a formulation capable of solving generative modeling tasks in higher dimensions, and motivated by the success of diffusion-based and flow-matching approaches, we propose a *dynamic*, time-dependent extension of the functional over the interval $t \in [0, T]$. Let $F(x, t)$ be the dynamic flux, $\rho(x, t)$ be the density, and $v(x, t)$ be the velocity. With $F = \rho_t v_t$, our dynamic functional is defined as:

$$M_\lambda^\alpha(F) = \inf_F \int_0^T \left( \lambda^{\gamma_1} \int_\Omega |F(x,t)|^\alpha \, dx + \lambda^{\gamma_2} \int_\Omega |\nabla F(x,t)|^2 \, dx \right) dt, \quad \text{s.t. Eq. 1} \qquad (5)$$

This dynamic formulation is a relaxation of the static one, often leading to a strictly lower energy. The core idea is that the additional time dimension allows for greater flexibility in how mass is transported. We formalize this advantage in the following informal lemma:

**Lemma 1:** (Dynamic Cost Reduction (Informal)) *The cost of any time-independent (static) transport strategy provides an upper bound for the dynamic problem. Due to the concavity of the transport term ($|F|^\alpha$ with $\alpha < 1$), concentrating the mass flow over a shorter time interval reduces its cost. Consequently, the two formulations coincide only if the optimal strategy is time-independent.*

This relationship is formally established in the Appendix D. This mechanism allows the model to automatically determine the optimal transport speed, balancing the incentive for fast, impulsive transport against the penalty for high spatial variations.

Although the dynamic functional (Eq. 15) provides a theoretically grounded relaxation, minimizing it directly is computationally intractable for high-dimensional generative modeling. The core issue lies in the flux formulation $F = \rho v$, which couples the velocity field with the instantaneous probability density. Minimizing the sublinear flux energy would require estimating $\rho(x, t)$ for every sample during training. To resolve this, we propose a velocity-driven objective that shifts the nonlinearity from the flux to the velocity field.

**Velocity-based objective.** We transform the objective into an expectation over the flow trajectory:

$$Y^\alpha(\rho, v) = \inf_{\rho, v} \underbrace{\int_0^1 \int_\Omega \rho_t |v_t|^\alpha \, dx dt}_{\mathcal{T}(\rho, v)} + \lambda \underbrace{\int_0^1 \int_\Omega \rho_t |\nabla v_t|^2 \, dx dt}_{\mathcal{C}(\rho, v)} \qquad (6)$$

This decoupling allows training via standard Neural ODE solvers by simply pushing particles and evaluating velocity norms, and not using exponents $\gamma$. This formulation is not equal to the flux-based approach, but drives the emergence of branching structures through two similar mechanisms.

**Why does this formulation produces Y-shaped flows?** The emergence of Y-shaped structures in this formulation arises from the competitive interplay between the sub-linear transport cost $\mathcal{T}$ and the cohesion regularizer $\mathcal{C}$. The optimal trajectory is determined by minimizing the total energy $Y^\alpha(\rho, v)$, which balances the geometric efficiency of fast, direct paths against the penalties incurred for breaking collective motion.

We begin by examining the behavior of the *cohesion* term $\mathcal{C}$, which acts as a penalty on spatial variations in the velocity field. A spatially uniform velocity field incurs a strictly zero cohesion

cost. This property provides a strong incentive for the mass to travel as a single unified cluster, referred to as the *Trunk* for an initial duration $t \in [0, \tau)$. Consequently, the system completely avoids cohesion penalties during this initial phase of transport. We formalize this observation in the following proposition:

**Proposition 1:** (The Zero-Cost of Rigid Translation) *If the velocity field $v(x, t)$ represents a spatially uniform translation (rigid body motion) on the support of $\rho$, the Cohesion Cost density is zero.*

To satisfy the boundary conditions, the mass must eventually separate to reach the distinct targets $y_1$ and $y_2$. This separation forces the velocity field to diverge, transitioning from a uniform vector to opposing directions over the small support width $\epsilon$. Our functional creates a localized spike in the energy functional proportional to $1/\epsilon$. This high cost acts as an activation barrier, discouraging early or gradual separation and instead favoring a delayed branching event. We formalize this as follows:

**Proposition 2:** (The High Cost of Spatial Velocity Conflict) *For a continuous density $\rho$ with connected support, if the velocity field $v$ attempts to separate the mass into distinct directions, the Cohesion Cost is strictly positive and scales inversely with the support width $\epsilon$.*

The feasibility of delaying the split is uniquely enabled by the sub-linear transport exponent $\alpha \in (0, 1)$. In standard quadratic Optimal Transport ($\alpha = 2$), rapid motion is prohibitively expensive. However, for $\alpha < 1$, the transport cost scales as $T^{1-\alpha}$. Consequently, traversing the distance over a reduced time interval $(1 - \tau)$ is energetically cheaper than traversing the distance over the full interval. This *time-compression* effect effectively discounts the geometric detour required by the branches. We formalize this kinetic property in the following proposition:

**Proposition 3:** (Time-Compression) *For $\alpha \in (0, 1)$, the transport cost of traveling a fixed distance $D$ decreases as the duration of the travel $T$ decreases. This favors "impulsive" motion (waiting and then bursting) over constant speed motion.*

In summary, by choosing a branching time $\tau^* > 0$, the system exploits the zero-cost dynamics of the trunk described in **Proposition 1** while simultaneously leveraging the sub-linear transport dynamics of **Proposition 3** to minimize the cost of the subsequent branches. The branching point represents the optimal trade-off where the savings derived from collective motion and time-compressed travel outweigh the geometric penalty of the path and the fixed cost of the split described in **Proposition 2**. Based on these properties, we propose the following lemma:

**Lemma 2:** (Existence of Optimal Branching Time) *For a sufficiently large cohesion weight $\lambda$ and $\alpha \in (0, 1)$, the optimal branching time $\tau^*$ satisfies $\tau^* > 0$, which implies that a Y-shaped trajectory is energetically superior to a V-shaped trajectory ($\tau = 0$).*

For the full proof of each proposition and lemma, please refer to Appendix E.

## 5 PRACTICAL IMPLEMENTATION

We now describe a simple and scalable training scheme based on neural ODEs and a Monte Carlo (MC) approximation of the velocity-power action, now augmented with the cohesion regularization term. Throughout this section, we fix the time horizon to $[0, 1]$.

We parameterize the velocity field by a neural network $v_\theta : \Omega \times [0, 1] \to \mathbb{R}^d$ and evolve samples via the ODE 2: $\frac{dx_t}{dt} = v_\theta(x_t, t)$, $x_{t=0} \sim \mu_0$. Let $\Phi_t$ denote the flow map induced by Equation **??**. The time-$t$ distribution of particles is the pushforward $\rho_t = (\Phi_t)_\# \mu_0$; i.e., if $x_0 \sim \mu_0$ then $x_t = \Phi_t(x_0)$ is a draw from $\rho_t$. This simple fact justifies using equal-weight particles to approximate expectations under $\rho_t$; no importance weights are introduced or needed.

**Objective.** The total action we minimize combines the sub-linear transport cost with the cohesion regularizer, as motivated in Section 4. We minimize:

$$Y^\alpha(\theta) = \int_0^1 \mathbb{E}_{x \sim \rho_t} \left[ |v_\theta(x, t)|_2^\alpha + \lambda |\nabla_x v_\theta(x, t)|^2 \right] dt, \tag{7}$$

where $\alpha \in (0, 1)$ is the branching exponent and $\lambda > 0$ is the cohesion weight matching the method formulation. The term $\|\nabla_x v_\theta\|_F^2$ represents the Dirichlet energy of the velocity field, which is efficiently computed using automatic differentiation.

**Time discretization and MC estimator.** Let $0 = t_0 < t_1 < \cdots < t_K = 1$ be a time grid with steps $\Delta t_k = t_k - t_{k-1}$ (in the uniform case, $\Delta t_k \equiv \Delta t = 1/K$). For a minibatch $\{x_0^{(i)}\}_{i=1}^N \sim \mu_0$, we integrate our neural ODE along the grid to obtain states $x_{t_k}^{(i)} = \Phi_{t_k}(x_0^{(i)})$. A consistent estimator of Equation 7 is then:

$$\hat{Y}^\alpha(\theta) = \frac{1}{N} \sum_{i=1}^N \sum_{k=1}^K \Delta t_k \left( |v_\theta(x_{t_{k-1}}^{(i)}, t_{k-1})|_2^\alpha + \lambda |\nabla_x v_\theta(x_{t_{k-1}}^{(i)}, t_{k-1})|^2 \right). \tag{8}$$

In the common uniform grid case, this simplifies the outer time summation to $\frac{1}{K} \sum_k$. If an adaptive ODE solver is used, we evaluate on its internal time points and accumulate the corresponding $\Delta t_k$. Under exact integration, independence of $\{x_0^{(i)}\}$, and mesh refinement ($N, K \to \infty$), $\hat{Y}^\alpha(\theta) \to Y^\alpha(\theta)$ by the law of large numbers and Riemann sums.

**Boundary Matching.** To enforce $\rho_{t=1} = \mu_1$, we add a differentiable boundary loss that compares the empirical law of the final particles to $\mu_1$. Given mini-batches $X_1 = \{x_{t=1}^{(i)}\}_{i=1}^{N_X}$ (pushforward samples) and $Y = \{y^{(j)}\}_{j=1}^{N_Y} \sim \mu_1$, we use the (bias-corrected) entropic Sinkhorn divergence Cuturi (2013); Feydy et al. (2019), with cost $c(x, y) = \|x - y\|_2^2$. Where $OT_\epsilon$ is the entropic OT cost computed by Sinkhorn iterations (regularization $\epsilon > 0$).

**Final Loss.** Our total training objective is:

$$\mathcal{L}(\theta) = \hat{Y}^\alpha(\theta) + \lambda_{sink} \mathcal{L}_{sinkhorn}(\theta), \tag{9}$$

where $\lambda_{sink} > 0$ balances the boundary matching constraint against the trajectory action. We differentiate through the ODE solver (using standard backpropagation or the adjoint method) and optimize $\theta$ with stochastic gradient methods using fresh mini-batches from $\mu_0$ and $\mu_1$. In practice, for efficient Jacobian computation, Hutchinson trick was applied Grathwohl et al. (2018).

## 6  RELATED WORK

Branching transport problems were studied in discrete settings. Classical approaches include the Euclidean Steiner Tree (ESTP), where branch points are optimized geometrically under flow-independent costs, and early geometric constructions for single-source branched optimal transport (BOT) (Buttazzo & Stepanov, 2003; Bernot et al., 2005; Maddalena et al., 2003). More recently, (Lippmann et al., 2022) introduced an approximate solver that decouples topology search from geometry optimization: given a tree topology, branch point positions are optimized via leaf-elimination in $O(nd)$ time, while a edge-rewiring procedure explores the topology. Their method works only for *discrete* settings.

Beyond discrete graph solvers, continuous stochastic frameworks have also been proposed. (Tang et al., 2025) extend Schrödinger Bridge Matching (BSBM) to the *branched* case by learning diverging drift and growth fields that transport mass from a common source to multiple targets. This method produces continuous branched trajectories, but is not based on branched transport theory.

## 7  EXPERIMENTS

### 7.1  BRANCHING ILLUSTRATIONS

**Gaussian Mixtures**. We compare how two flow-based methods transport mass from a single source distribution to a highly multi-modal target. The target is a mixture of $K$ clusters ("branches") arranged at the top; the source is a single cluster at the bottom. We train the same-size models with (i) standard FM and (ii) Y-Flows. Each curve shows the trajectory of a sample from source (light) to destination cluster (dark). Both models use the same time-conditioned MLP $v_\theta(x, t)$: 64-$d$ time embedding $\to$ 3×256 SiLU layers $\to$ 2-D output; integrated with 10 fixed Euler steps over $t \in [0, 1]$. Training is identical: Adam ($lr = 10^{-3}$, batch 256, 10k iterations, seed 42. FM uses the standard flow-matching objective along linear source–target couplings. Y-Flows maintains the same backbone, but increases the loss with a branched-transport prior: endpoint OT/Sinkhorn term weight $\lambda_{sink} \approx 5$. For the results, see Figure 2. As shown, Y-Flows discovers a shared *trunk* that later splits into branches, producing short, structured, tree-like transport.

To leverage the fast convergence properties, we propose an early stopping mechanism for the neural ODE solver. Specifically, we terminate the integration if the displacement norm between consecutive

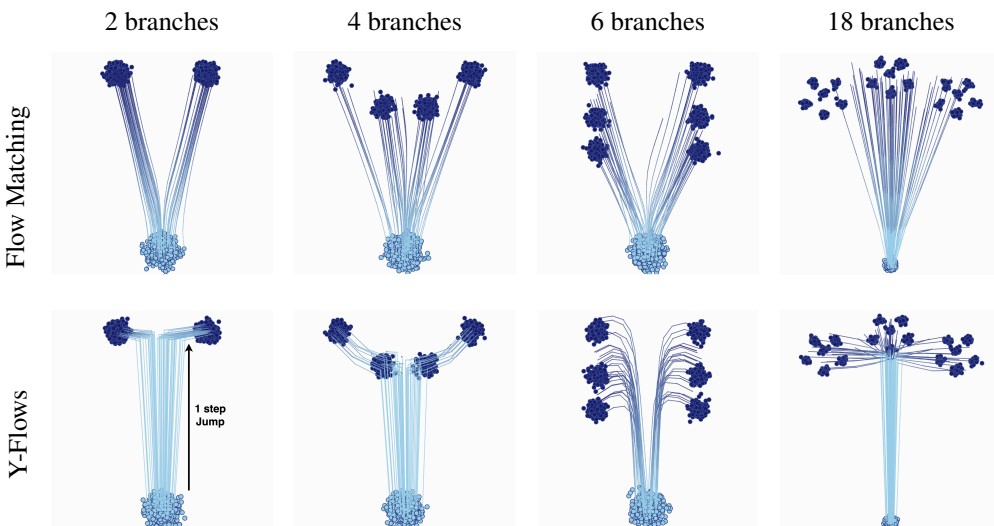

Figure 2: Comparison of Gaussian mixture toy tasks. Each **column** is a task (T-shape, 4,6,18 Branches). Each **row** is a method: top = Flow Matching (FM), bottom = Y-Flows (ours). The color gradient represents the flow steps. A monotone color indicates that the number of steps on this region was equal to 1. As shown in the 2, 4, and 18-branch cases, our model initially made a significant jump toward the target before splitting the mass. Subsequent movement involved reaching the targets via small almost zero size steps.

steps, $\|x_{t+1} - x_t\|_2$, falls below a threshold of $\delta = 1 \times 10^{-3}$. This criterion effectively detects when particles have settled into their target modes, allowing the model to dynamically adapt its computational budget. Empirically, we observe that on the Gaussian mixtures, Y-Flows converged in an average of 3 steps for the 2-branch case, 5 steps for 4 branches, 10 steps for 6 branches, and 4.5 steps for the 18-branch setting. In contrast, the Flow Matching baseline consistently utilized a fixed budget of 10 integration steps to achieve comparable transport, highlighting our method's ability to concentrate mass movement into fewer updates.

**LiDAR Surface Navigation**. The purpose of this experiment is to test whether a branched OT model *can learn when and where to split while staying confined to a real 3D surface*. For this experiment, we include a potential energy penalty on our loss as in (Liu et al., 2023; Tang et al., 2025). We transport a single source to four target clouds laid out on an airborne LiDAR terrain ($\sim$ 5k normalized points after ground/low-veg filtering, see F.1), which adds curvature, uneven sampling, and natural surface corridors that synthetic mixtures do not capture. The method infers a three-junction topology, places the splits at terrain transitions (ridge breaks/valley entries), and keeps trajectories on-surface, reconstructing all four targets with tight endpoint fits. This shows that the approach handles surface-constrained, branched transport and is applicable in real-world tasks; see Figure 3 for details.

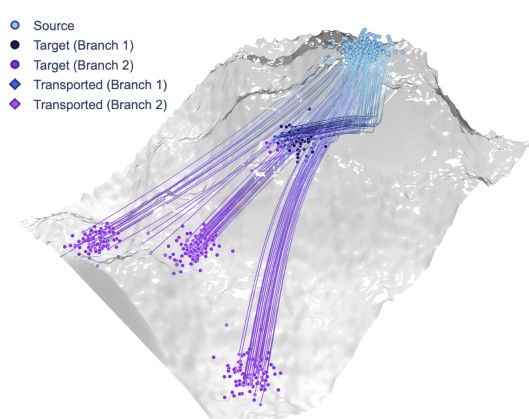

Figure 3: *Result of Y-Flows on LiDAR dataset.*

## 7.2 BIOLOGY DATA

**Cellular Differentiation**. To further validate our method, we use the Tedsim dataset (Pan et al. (2022)) as a controlled reference point with known ground truth dynamics. Tedsim simulates a cellular differentiation process by modeling cell division from a root cell, generating both gene expression profiles and heritable lineage barcodes. This provides a complete record of a simple, branching differentiation process.

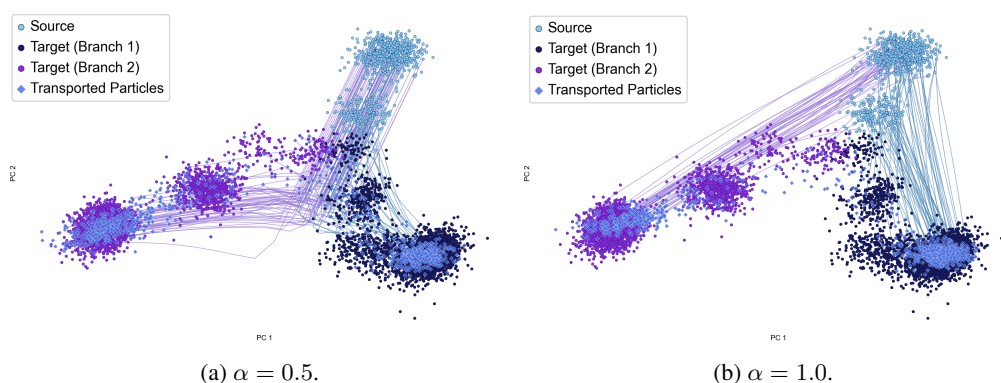

(a) $\alpha = 0.5$.
(b) $\alpha = 1.0$.

Figure 4: *PCA projected results of Y-Flows on Tedsim dataset (50D).*

Table 1: Comparison on **Tedsim** and **Single-Cell RNA** at 250D embedding. Lower-is-better, best in **bold**.

| Dataset | Metric | Method | | | | |
|---|---|---|---|---|---|---|
| | | BSBM | CNF | CFM | FM | Y-Flows |
| Tedsim | $W_1$ | 23.47 | 22.48 | 20.07 | 22.80 | **17.39** |
| | $W_2$ | 23.71 | 22.51 | 20.14 | 22.94 | **17.60** |
| | MMD | 0.55 | 0.18 | 0.12 | 0.16 | **0.11** |
| Single-Cell RNA | $W_1$ | 30.36 | 22.13 | 17.67 | 18.07 | **16.37** |
| | $W_2$ | 30.77 | 22.49 | 17.95 | 18.36 | **16.26** |
| | MMD | 0.31 | 0.15 | **0.11** | 0.12 | **0.11** |

We apply our method to a specific Tedsim scenario, modeling the transport from a single progenitor cell to two distinct terminal states. This controlled experiment allows us to quantitatively assess our method's ability to accurately reconstruct branching trajectories where the true pathways are known beforehand. See Figure 4 and Table 1. The experiments highlight our model's effectiveness in reconstructing branching trajectories. We observed that sub-linearity exponents of $\alpha = 0.5$ consistently produced well-defined bifurcating structures. We compared different approaches using the following metrics: W1/W2: Wasserstein distance with ground metrics, see Section 3. RBF-MMD: Maximum Mean Discrepancy using a Radial Basis Function kernel with a median heuristic bandwidthGretton et al. (2012). See Table 3 for results in higher dimensions.

**Single-cell RNA**. We now evaluate the performance of our method on the 50D Paul15 dataset. This dataset (Paul et al., 2015) contains single-cell RNA sequencing data from approximately 2,730 myeloid progenitor cells, with expression profiles for roughly 1,000 highly variable genes. The dataset captures various cell types in the hematopoietic system, including early progenitor cells, monocytes, and neutrophils. This dataset represents cellular states during myeloid differentiation and serves as a benchmark for studying developmental trajectories and cell fate decisions in hematopoiesis. We model transport from early progenitor cells to mature, differentiated cell types (monocytes and neutrophils). For comparison with other methods such as Branched SBM (Tang et al., 2025) and FM (Lipman et al., 2022), see Table 1. For results in higher dimensions, see Table 4.

## 8 IMAGE DATA

**FFHQ dataset.** To test our model in higher dimensions we run experiments in 512 D latent space of a pretrained ALAE model on FFHQ 1024x1024 dataset. The ALAE is used *only* for decoding latents to images during evaluation and remains frozen at all times. We load latent vectors paired with gender labels and split them into 60k training and 10k test samples. Let $\mathcal{D}$ denote the latent dimensionality inferred from the data.

**Settings**: *Unconditional generation*: learn a vector field that transports samples from a standard Gaussian prior $\mu_0 \sim \mathcal{N}(0, I_\mathcal{D})$ to the empirical distribution of data latents $\mu_1$. **2)** *Domain translation*: learn a class-conditional transport from source-class latents to target-class latents (etc. female $\rightarrow$ male); mini-batches for $\mu_0$ and $\mu_1$ are sampled from the respective class-specific training subsets.

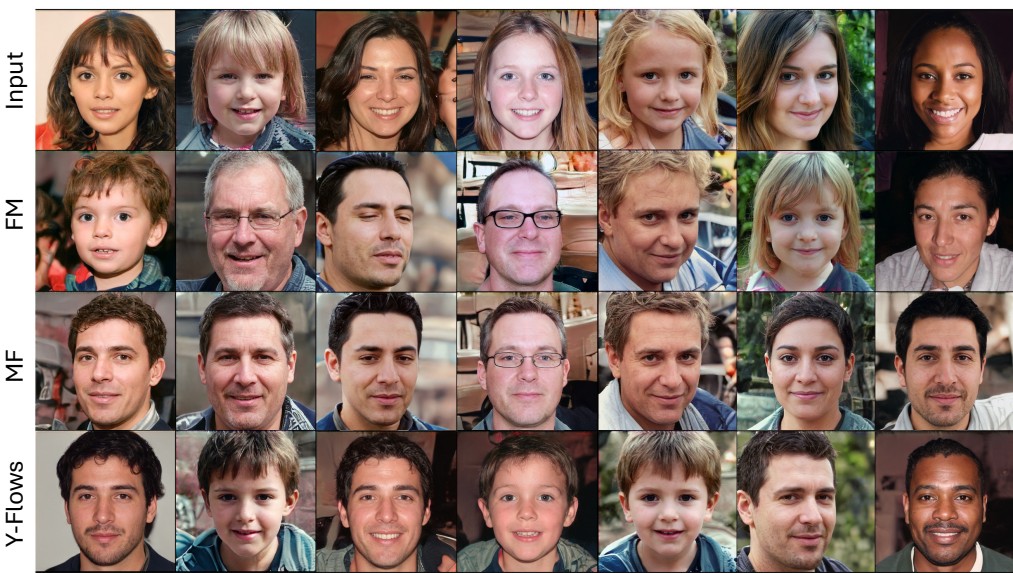

Figure 5: *Female-to-male domain translation in the ALAE latent space. Two-step ODE results. Our method produces compatible results in a higher-dimensional latent space, as do other generative models.*

**Architecture**: $v_\theta$ is an MLP with Tanh activations and hidden width 1024. A scalar time embedding (linear layer) is concatenated to $x$ before the MLP. We use the same architecture for fair comparison for all types of model except for Mean Flows which requires additional input $r$ to encode the start of sampling trajectory.

**Baselines**: We compare against Flow Matching (FM) and Mean Flows (MF) (Geng et al., 2025). All methods are trained for 100k iterations with a learning rate of $10^{-4}$ using the Adam optimizer.

**Results:** Figure 5 visualizes the domain translation results. Top row: samples from the source distribution $\mu_0$ (female faces). Subsequent rows show the corresponding translations into the target distribution $\mu_1$ (male faces) using FM, MF, and Y-Flows. Columns share the same source sample for cross-method comparison. Qualitatively, Y-Flows better preserves key attributes such as apparent age and skin tone. Quantitatively, we report Fréchet Distance (**FD**) on FFHQ latents. As shown in Table 2, Y-Flows attains

Table 2: Fréchet distance **val** vs. steps by model (↓) on FFHQ latents.

| Steps | FM | MF | Y-Flows |
|---|---|---|---|
| 1 | 35.17 | 35.87 | 35.27 |
| 2 | 26.33 | 23.56 | **23.45** |

the best scores and achieves high-quality translations in as few as two steps. See Figure 8 for the unconditional generation setup. This justifies our *time-compression* theoretical assumption on our method.

**Run-time:** All experiments were conducted on a single NVIDIA RTX A6000 GPU. The average training time per experiment takes $\sim$ 1 hour. To *reproduce* our experiments, refer to the supplementary materials. The code will be open-source. Details on the hyper-parameters used are presented above.

## 9 CONCLUSION

In this work, we introduced Y-shaped generative flows, a continuous-time framework that addresses the structural limitations of standard V-shaped transport by rewarding shared movement before branching. By minimizing a novel velocity-power action with a sublinear exponent $\alpha \in (0, 1)$, we established a theoretical link between scalable neural ODEs and branched optimal transport. We demonstrated that this concave velocity dependence induces a time-compression effect, favoring fast, shared transport along trunks followed by efficient branching. Empirically, our method recovers interpretable lineage structures in biological data, respects geometric constraints in LiDAR navigation, and achieves high-quality image generation with significantly fewer integration steps. Ultimately, Y-Flows offer a theoretically grounded and practically effective approach for learning hierarchy-aware generative trajectories that adapt computational effort to the underlying data structure.

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

## A BACKGROUND (EXTENDED)

**Continuous Normalizing Flow** A Continuous Normalizing Flow (CNF) Chen et al. (2018) is a generative model that defines a probability density path through a Neural Ordinary Differential Equation (ODE):

$$\frac{d}{dt}x_t = v_\theta(x_t, t), \qquad x_{t=0} \sim \mu_0.$$

Let $\Phi_t$ be the flow map associated with this ODE, which transports a particle from its initial condition at time 0 to its location at time $t$. The pushforward density $\rho_t = (\Phi_t)_\# \rho_0$ evolving under this dynamics *necessarily* satisfies the *continuity equation* with the parameterized velocity field $v_\theta$. The continuity equation encodes the law of mass conservation:

$$\partial_t \rho_t + \nabla \cdot (\rho_t v_t) = 0 \quad \text{on } \Omega \times (0, 1),$$
$$\rho_{t=0} = \rho_0, \quad \rho_{t=1} = \rho_1.$$

A key result is the instantaneous change of variables formula, which describes how the log density evolves along a trajectory:

$$\frac{d}{dt} \log \rho_t(x_t) = -\nabla \cdot v_\theta(x_t, t).$$

This allows for a likelihood calculation by integrating this quantity over time. Training can be done by directly maximizing likelihood (integrating equation A).

**Flow Matching.** The core idea of Flow Matching (FM)(Lipman et al., 2022; Tong et al., 2023) is to train a CNF by directly regressing its velocity field $v_\theta$ toward a target vector field $u_t$ that generates a desired probability path. The Flow Matching objective is: $\mathcal{L}_{\text{FM}}(\theta) = \mathbb{E}_{t \sim \mathcal{U}[0,1], \, x \sim \rho_t} \left[ \|v_\theta(x, t) - u_t(x)\|^2 \right]$.

A critical challenge is that sampling $x \sim \rho_t$ from the marginal path at arbitrary times is typically intractable. Conditional Flow Matching (CFM) (Lipman et al., 2022) provides a solution by constructing the marginal path as a mixture of simpler and tractable conditional paths. Let $z$ be a conditioning variable with distribution $q(z)$. We define the marginal path as follows: $\rho_t(x) = \int \rho_t(x \mid z)q(z) \, dz$, where each conditional path $\rho_t(x|z)$ is generated by a corresponding conditional vector field $u_t(x \mid z)$. The marginal field $u_t(x)$ that generates $\rho_t$ is then given by:

$$u_t(x) = \mathbb{E}_{z \sim q(z|x)}[u_t(x \mid z)] = \mathbb{E}_{z \sim q} \left[ \frac{\rho_t(x \mid z)}{\rho_t(x)} u_t(x \mid z) \right],$$

where $q(z \mid x)$ is the posterior. The key theorem is to minimize the following *Conditional Flow Matching* objective:

$$\mathcal{L}_{\text{CFM}}(\theta) = \mathbb{E}_{t \sim \mathcal{U}[0,1], \, z \sim q, \, x \sim \rho_t(\cdot|z)} \left[ \|v_\theta(x, t) - u_t(x \mid z)\|^2 \right] \tag{10}$$

yields the same gradient for $\theta$ as minimizing the intractable $\mathcal{L}_{\text{FM}}(\theta)$. This makes CFM a practical objective, as it only requires sampling from conditional paths $\rho_t(x|z)$ and knowing their closed-form drifts $u_t(x|z)$.

The flexibility of CFM lies in the choice of conditional paths. The coupling $q(z)$ is the independent joint distribution $q(x_0)q(x_1)$, so $z = (x_0, x_1)$. A common conditional path is a Gaussian bridge: $\rho_t(x \mid z) = \mathcal{N}(x \mid \mu_t, \sigma^2 I)$, where $\mu_t = (1-t)x_0 + tx_1$ is a linear interpolation. The simple constant drift that generates this path is $u_t(x \mid z) = x_1 - x_0$. The optimal transport CFM (OT-CFM) (Tong et al., 2023) method uses an optimal coupling to define conditionals. Here, $z = (x_0, x_1)$ is sampled from an OT plan $\pi$ between $\mu_0$ and $\mu_1$ 2. In practice, the OT plan $\pi$ is efficiently approximated using mini-batch OT, which has been shown to work well empirically.

## B MODICA-MORTOLA BRANCHED FLOWS

Oudet & Santambrogio (2011) proposed a theoretically grounded approximation inspired by the Modica-Mortola framework used in phase-field modeling. This framework replaces the singular branched transport problem with a sequence of regularized, elliptic energy functionals $M_\lambda^\alpha$, defined

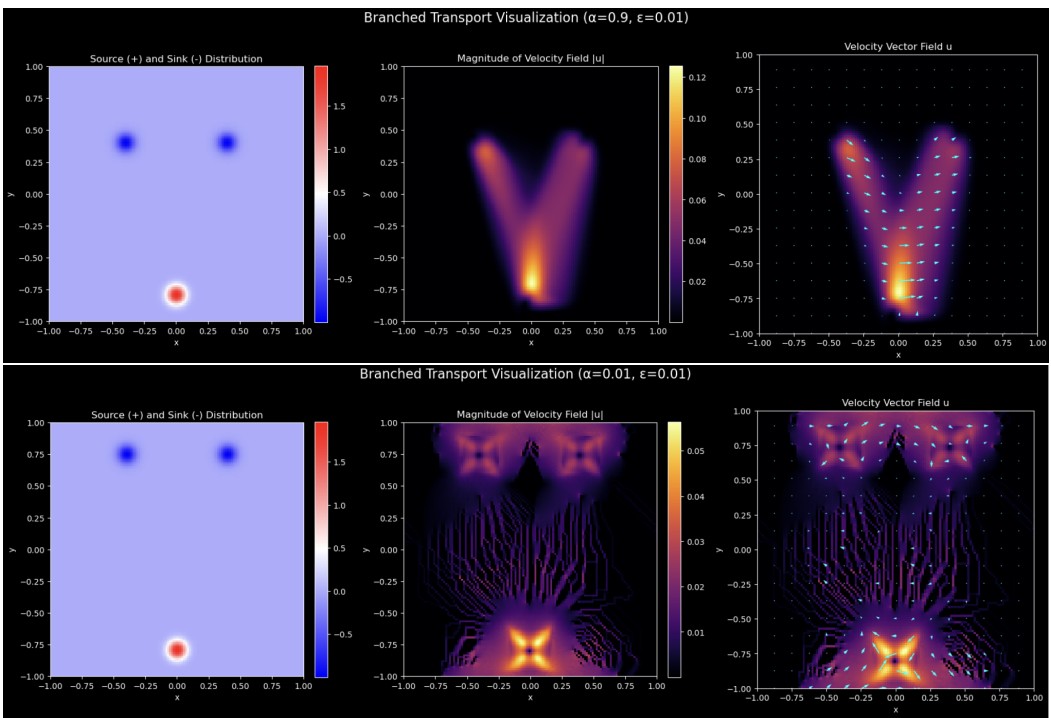

Figure 6: Modica-Mortola solver results using a neural network-based field parametrization. The difference between the top and bottom rows illustrates the sensitivity of the method to optimization parameters, specifically the learning rate. We observe that the method is highly sensitive to initialization and optimization settings.

over the more regular space of $\mathcal{H}^1$ vector fields. For a given vector field $F(x)$, the approximating functional is defined as:

$$M_\lambda^\alpha(F) = \lambda^{\gamma_1} \int_\Omega |F(x)|^\alpha dx + \lambda^{\gamma_2} \int_\Omega |\nabla F(x)|^2 dx, \qquad (11)$$

where $\lambda > 0$ is a small regularization parameter. The exponents $\gamma_1$ and $\gamma_2$ are derived from the transport dimension $d$ and the cost exponent $\alpha$ to ensure correct scaling behavior. As described by Oudet & Santambrogio (2011), the functional in Eq. 11 consists of two competing terms:

1. **A concave potential term** ($\int |F|^\alpha dx$): This term acts analogously to a double-well potential, encouraging the magnitude of the vector field to be either zero or arbitrarily large, thereby promoting the formation of sparse, high-intensity pathways.

2. **A Dirichlet energy term** ($\int |\nabla F|^2 dx$): This Sobolev penalty penalizes spatial variations, enforcing smoothness and regularity on the vector field.

The central theoretical guarantee is that as $\lambda \to 0$, the functionals $M_\lambda^\alpha$ $\Gamma$-converge to the original branched transport energy $M^\alpha$. This ensures that minimizers of the regularized problem converge to a minimizer of the intractable branched transport problem.

While the original implementation by Oudet & Santambrogio (2011) relied on a staggered grid discretization and FFT-based projections to enforce the divergence constraint, we reformulated this variational principle within a deep learning framework.

In our experiments, we parameterized the vector field $F$ as a coordinate-based neural network (an MLP) $F_\theta : \Omega \to \mathbb{R}^d$, implemented in PyTorch. Unlike grid-based methods, this allows for a mesh-free representation of the transport density. The optimization objective $\mathcal{L}(\theta)$ is composed of the relaxed energy functional and a soft penalty for the divergence constraint:

$$\mathcal{L}(\theta) = M_\lambda^\alpha(F_\theta) + \beta \|\nabla \cdot F_\theta - (\mu_0 - \mu_1)\|^2, \qquad (12)$$

where $\beta$ is a penalty weight (In practice, we find it important to put this value high, in our experiments, the method was working only with $\beta > 10k$). We utilized PyTorch's automatic differentiation

(`torch.autograd`) to compute the spatial Jacobian $\nabla F_\theta$ for the Dirichlet term and the divergence $\nabla \cdot F_\theta$ for the constraint term exact to the network precision. The integrals were approximated via Monte Carlo sampling over the domain $\Omega$.

We evaluated this approach on a simple setting with a single source and two symmetric targets. While the method successfully generated the branching structures shown in Figure 6, we found the training dynamics to be notoriously difficult to stabilize. The optimization landscape is highly sensitive to hyperparameters; for instance, training diverged significantly at a learning rate of 1e-4, yet converged to a reasonable solution at 1e-5.

We attribute this difficulty to the inherent nature of the flux variable in the Modica-Mortola formulation. Since $F$ represents a mass flux (density $\times$ velocity), optimal solutions tend toward singularity (Dirac structures) as $\lambda \to 0$. Neural networks often struggle with such high-frequency spectral components (spectral bias), making the simultaneous minimization of the concave potential and the divergence constraint numerically unstable without careful scheduling of the regularization parameter $\lambda$.

## C   PROOFS

## D   RELATION BETWEEN STATIC AND DYNAMIC FORMULATIONS

We compare the proposed dynamic formulation with the static Modica–Mortola approximation.

**Static Formulation.** Consider the static approximating functional $E_\lambda$ defined on vector fields $u \in H^1(\Omega; \mathbb{R}^d)$:

$$E_\lambda(F) = \lambda^{\gamma_1} \int_\Omega |F(x)|^\alpha \, dx + \lambda^{\gamma_2} \int_\Omega |\nabla F(x)|^2 \, dx, \qquad (13)$$

where $\gamma_1 < 0 < \gamma_2$ are scaling exponents and $\alpha$ is the concave power (related to the transport exponent $\alpha$). This energy is subject to the divergence constraint accounting for source and sink distributions $f = f^+ - f^-$:

$$\nabla \cdot F = f \quad \text{in } \Omega, \qquad F \cdot n = 0 \text{ on } \partial\Omega. \qquad (14)$$

**Dynamic Formulation.** We consider a time-dependent extension over the interval $t \in [0, T]$. Let $F(x, t)$ be the dynamic flux. The dynamic energy is the time integral of the static spatial integrand:

$$\mathcal{J}_\lambda(u) = \int_0^T \left( \lambda^{\gamma_1} \int_\Omega |F(x,t)|^\alpha \, dx + \lambda^{\gamma_2} \int_\Omega |\nabla F(x,t)|^2 \, dx \right) dt, \qquad (15)$$

subject to the continuity equation $\partial_t \rho + \nabla \cdot F = 0$ with boundary conditions $\rho(0) = f^+$ and $\rho(T) = f^-$.

As shown in Lemma 1, the dynamic formulation is a relaxation of the static problem.

- If the dynamic flux is constant in time, the dynamic energy is proportional to the static energy (scaled by $T$).
- Due to the concavity of the term $|F|^\alpha$ (where typically $\alpha < 1$ for branched transport), the transport term obeys a "Time Compression" property: performing the same displacement in a shorter time interval strictly reduces the transport cost. This allows the dynamic cost to be strictly lower than the steady static solution.

Thus, the dynamic and static formulations coincide only when the optimal transport strategy is time-independent.

**Lemma 1** (Dynamic–Static Comparison with Modica–Mortola Scaling). *Let $\Omega$ be a bounded domain, $T > 0$, and parameters $\lambda, \gamma_1, \gamma_2, \alpha$ be fixed. Let $E_\lambda(m)$ be the static energy subject to $\nabla \cdot m = f$, and $\mathcal{J}_\lambda(F)$ be the dynamic energy subject to $\partial_t \rho + \nabla \cdot F = 0$.*

*(i) **Upper Bound:** The dynamic infimum is bounded by the static energy of a steady competitor:*

$$\inf_{\rho, F} \mathcal{J}_\lambda(F) \leq T \cdot E_\lambda(m/T) = \lambda^{\gamma_1} T^{1-\alpha} \int_\Omega |m|^\alpha \, dx + \lambda^{\gamma_2} T^{-1} \int_\Omega |\nabla m|^2 \, dx,$$

*where $m$ is any static field satisfying $\nabla \cdot m = f$.*

*(ii)* ***Strict Inequality via Time Compression:*** *If $\alpha < 1$, time compression of a feasible trajectory can yield an energy strictly lower than the steady-state bound.*

*Proof.* **(i) Reduction to static constraint.** Consider a time-independent flux $F(x,t) = \frac{1}{T}m(x)$ where $\nabla \cdot m = f$. Integrating the continuity equation over $[0,T]$ confirms this flux satisfies the mass transport boundary data. Substituting $F \equiv m/T$ into the dynamic energy:

$$\mathcal{J}_\lambda(F) = \int_0^T \left( \lambda^{\gamma_1} \int_\Omega \left|\frac{m}{T}\right|^\alpha dx + \lambda^{\gamma_2} \int_\Omega \left|\nabla\frac{m}{T}\right|^2 dx \right) dt$$

$$= T\left( \lambda^{\gamma_1} T^{-\alpha} \int_\Omega |m|^\alpha dx + \lambda^{\gamma_2} T^{-2} \int_\Omega |\nabla m|^2 dx \right),$$

which yields the bound in statement (i).

**(ii) Time compression.** Let $F(x,t)$ be any feasible dynamic flux on $[0,T]$. For a compression factor $s \in (0,1)$, define the compressed flux $F_s(x,t) = \frac{1}{s}F(x,t/s)$ on the interval $[0,sT]$. It is straightforward to verify that $F_s$ satisfies the continuity equation with the same boundary data. Substituting $F_s$ into the energy functional and changing variables $\tau = t/s$ yields the scaling law:

$$\mathcal{J}_\lambda(F_s) = s^{1-\alpha}\mathcal{T}(F) + s^{-1}\mathcal{D}(F),$$

where $\mathcal{T}(F) = \int_0^T \lambda^{\gamma_1} \int_\Omega |F|^\alpha d\tau$ is the transport energy and $\mathcal{D}(F) = \int_0^T \lambda^{\gamma_2} \int_\Omega |\nabla F|^2 d\tau$ is the dissipative energy. Since $\alpha < 1$, the exponent $1 - \alpha$ is positive. Thus, the transport cost decreases as $s \to 0$ (favoring fast transport), while the dissipative cost increases as $s^{-1}$ (penalizing speed). Minimizing this function with respect to $s$ yields a finite optimal time scale $s_*$. If the parameters satisfy $s_* < 1$, then $\mathcal{J}_\lambda(F_{s_*}) < \mathcal{J}_\lambda(F)$, proving that the dynamic minimum can be strictly lower than the uncompressed (steady) state. $\square$

A key feature of the proposed dynamic formulation is its ability to automatically determine the optimal transport speed. This behavior arises from the structure of the energy functional, where the transport cost is sub-additive (concave power $\alpha < 1$). This mechanism actively drives the system to execute the mass transfer as rapidly as possible, rather than smearing it diffusively over the entire time horizon.

However, this drive for infinite speed is balanced by the Dirichlet regularization term ($\int |\nabla u|^2$), which scales inversely with time and acts as a penalty on excessive velocity. The competition between these two terms results in a finite, optimal duration for the transport event. This ensures that the model naturally recovers the sharpest possible transition profile permissible by the regularization $\lambda$, representing a distinct advantage over static formulations where the time profile is fixed or irrelevant.

# E    FORMALIZATION OF BRANCHING INTUITION

We consider a dynamic Optimal Transport problem on a domain $\Omega \subset \mathbb{R}^d$ over a time interval $t \in [0,1]$. The goal is to transport a source distribution $\rho_0$ to a target $\rho_1$ while minimizing a composite energy functional. Let $\Omega \subset \mathbb{R}^d$. We consider a dynamic formulation where density $\rho_t$ moves according to velocity $v_t$. The total energy functional is:

$$J(\rho, v) = \underbrace{\int_0^1 \int_\Omega \rho_t \|v_t\|^\alpha \, dxdt}_{\mathcal{T}(\rho,v)} + \lambda \underbrace{\int_0^1 \int_\Omega \rho_t \|\nabla v_t\|_F^2 \, dxdt}_{\mathcal{C}(\rho,v)} \tag{16}$$

with respect to Eq. 1. Where $\alpha \in (0,1)$ is the sub-linear transport exponent. $\epsilon > 0$ regulates the cohesion strength.

Lets consider the symmetric toy problem. We define source: a mass distribution concentrated at $P = (0,0)$ at $t = 0$. Targets: two symmetric targets $Q_1 = (-w, h)$ and $Q_2 = (w, h)$ at $t = 1$. Particle Model: we model the mass as a cloud of width $\epsilon$. Strategy Parameter: we define a branching time $\tau \in [0,1]$. For $t \in [0,\tau]$, the mass moves as one cluster (Trunk). For $t \in [\tau, 1]$, the mass splits and moves to targets (Branches).

To provide a rigorous proof, we must model the tradeoff between the **Transport Cost** (path length-/speed) and the **Cohesion Cost** (the Dirichlet energy of the velocity field weighted by density).

**Proposition 1** (The Zero-Cost of Rigid Translation). *If the velocity field $v(x,t)$ represents a spatially uniform translation (rigid body motion) on the support of $\rho$, the Cohesion Cost density is zero.*

*Proof.* Let the support of $\rho_t$ be denoted as $\text{supp}(\rho_t)$. Assume $v(x,t) = u(t)$ for all $x \in \text{supp}(\rho_t)$, where $u(t)$ is a vector dependent only on time (not space).

The Frobenius norm of the gradient is defined as:

$$\|\nabla v\|_F^2 = \sum_{i,j} \left(\frac{\partial v_i}{\partial x_j}\right)^2 \tag{17}$$

Since $v(x,t) = u(t)$ is constant with respect to spatial coordinates $x$:

$$\frac{\partial v_i}{\partial x_j} = 0 \quad \forall i,j \implies \int_{\text{supp}(\rho)} \rho\|\nabla v\|_F^2 \, dx = \int \rho \cdot 0 \, dx = 0 \tag{18}$$

$\square$

The *Trunk* phase of a Y-shape structure incurs zero penalty from the cohesion term.

**Proposition 2** (The High Cost of Spatial Velocity Conflict). *For a continuous density $\rho$ with connected support, if the velocity field $v$ attempts to separate mass into distinct directions (divergence/shear), the Cohesion Cost is strictly positive and scales inversely with the support width $\epsilon$.*

*Proof.* Consider a 1D cross-section of the particle cloud at the moment of splitting. Let $\rho(x)$ be defined on $[-\epsilon, \epsilon]$. To split the mass:

- $v(x) \approx -u$ for $x \in [-\epsilon, 0)$ (moving left toward $Q_1$)

- $v(x) \approx +u$ for $x \in (0, \epsilon]$ (moving right toward $Q_2$)

For $v$ to be differentiable, it must transition from $-u$ to $+u$ over the distance $2\epsilon$. By the Mean Value Theorem, there exists some $x_0$ where $|\nabla v(x_0)| \geq \frac{u}{\epsilon}$. The contribution to the cost is:

$$\int \rho\|\nabla v\|^2 dx \approx \rho_{avg}\left(\frac{u}{\epsilon}\right)^2 \cdot (2\epsilon) \propto \frac{1}{\epsilon} \tag{19}$$

$\square$

**Proposition 3** (Time-Compression). *For $\alpha \in (0,1)$, the transport cost of traveling a fixed distance $D$ decreases as the travel duration $T$ decreases. This favors "impulsive" motion (waiting and then bursting) over constant speed motion.*

*Proof.* Let a particle travel distance $D$ over duration $T$ with constant speed $v = D/T$. The Transport Cost is:

$$C(T) = \int_0^T |v|^\alpha dt = T \cdot \left(\frac{D}{T}\right)^\alpha = D^\alpha T^{1-\alpha} \tag{20}$$

We analyze the behavior of $C(T)$ with respect to $T$:

$$\frac{dC}{dT} = D^\alpha(1-\alpha)T^{-\alpha} \tag{21}$$

Since $\alpha \in (0,1)$, we have $(1-\alpha) > 0$. Thus, $\frac{dC}{dT} > 0$. $\square$

In a Y-shape, the *Branches* must traverse the distance to the targets in the reduced time window $(1-\tau)$. In standard Optimal Transport ($\alpha = 2$), this *rushed* travel is expensive ($Cost \propto 1/T$). However, with $\alpha < 1$, the cost scales as $T^{1-\alpha}$. Reducing the time window actually **lowers** the transport cost of the branches. This effectively "discounts" the geometric penalty of the trunk, making branching even more favorable.

**Lemma 2** (Existence of Optimal Branching Time). *For a sufficiently large cohesion weight $\lambda$ and $\alpha \in (0, 1)$, the optimal branching time $\tau^*$ satisfies $\tau^* > 0$, implying a Y-shape trajectory is energetically superior to a V-shape trajectory ($\tau = 0$).*

*Proof.* Let the branching point $S$ be located at height $y = \tau h$. We minimize the total energy $E(\tau)$ with respect to $\tau \in [0, 1]$.

**1. Transport Cost ($\mathcal{T}$):** The total path length consists of the trunk (length $\tau h$, duration $\tau$) and two branches (length $L_2(\tau) = \sqrt{w^2 + h^2(1 - \tau)^2}$, duration $1 - \tau$). Using time-compression proposition:

$$\mathcal{T}(\tau) = (\tau h)^\alpha \tau^{1-\alpha} + 2 \cdot \frac{1}{2}(L_2(\tau))^\alpha (1 - \tau)^{1-\alpha} \tag{22}$$

$$\mathcal{T}(\tau) = h^\alpha \tau + \left(\sqrt{w^2 + h^2(1 - \tau)^2}\right)^\alpha (1 - \tau)^{1-\alpha} \tag{23}$$

**2. Cohesion Cost ($\mathcal{C}$):** From proposition 1, cost is 0 during the trunk. From The High Cost of Spatial Velocity Conflict proposition, cost is incurred during the split phase $[\tau, 1]$. Let $K_{\text{split}}$ be the cost rate.

$$\mathcal{C}(\tau) = K_{\text{split}}(1 - \tau) \tag{24}$$

**3. Derivative at $\tau = 0$:** We check if increasing $\tau$ from 0 reduces energy (i.e., is $E'(0) < 0$?).

$$E'(\tau) = \mathcal{T}'(\tau) + \mathcal{C}'(\tau)$$

**Transport Term:** Because of the $\alpha$-Dynamics (Lemma 3), the derivative of the branch cost contains a negative term from the shrinking time duration $(1 - \tau)$. The transport penalty for taking the longer Y-path is significantly dampened compared to the $\alpha \geq 1$ case. **Cohesion Term:** Since $\mathcal{C}(\tau)$ decreases linearly with $\tau$:

$$\mathcal{C}'(0) = -K_{\text{split}}$$

**Conclusion:** Combining terms, we have a geometric penalty (positive) and two incentives for branching (negative):

$$E'(0) = \text{Geometric Penalty} - \text{Time-Acceleration Bonus} - \lambda K_{\text{split}}$$

For sufficiently large $\lambda$ (or sufficiently small $\alpha$), the negative terms dominate:

$$E'(0) < 0$$

Thus, the system naturally evolves a **Y-shape** ($\tau^* > 0$) to exploit the zero-cost trunk and the low-cost high-speed branches. $\qquad\square$

## F ADDITIONAL EXPERIMENTS AND ILLUSTRATIONS

### F.1 LIDAR

Airborne LiDAR Liu et al. (2023) terrain tile with ground and low-vegetation; each sample is a 3D point $(x, y, z)$ with optional attributes (intensity, return number, total returns, scan angle, class). After filtering non-ground classes, removing outliers with a radius/k-NN check, and Poisson-disk (or voxel-grid) subsampling, we retain 5k points that preserve ridges, slopes, and basins. Coordinates are centered and scaled to a unit box (optionally PCA-aligned). We estimate per-point normals via local PCA and build a k-NN surface graph ($k \in [8, 16]$) with Euclidean edge weights as a geodesic proxy; during optimization, states are projected back to the nearest tangent patch to stay on-manifold. A single source distribution sits on a lower slope; four target clouds are disjoint regions on ridges/basins (indices provided for reproducibility).

### F.2 BIOLOGY DATA

### F.3 IMAGE DATA

The implementation of *Flow Matching* was based on:
`facebookresearch/flow_matching`.

Table 3: Comparison across methods on Tedsim dataset across embedding dimensions.

| | Metric | BSBM | CNF | CFM | FM | Y-Flows |
|---|---|---|---|---|---|---|
| | $W_1$ | 17.72 | 13.76 | 12.33 | 12.09 | **10.48** |
| 50D | $W_2$ | 17.96 | 13.81 | 12.44 | 12.17 | **10.72** |
| | RBF-MMD | 0.63 | 0.51 | 0.11 | 0.15 | **0.10** |
| | $W_1$ | 20.34 | 18.84 | 17.45 | 18.60 | **14.29** |
| 150D | $W_2$ | 20.51 | 18.88 | 17.54 | 19.79 | **14.42** |
| | RBF-MMD | 0.48 | 0.29 | 0.13 | 0.14 | **0.13** |
| | $W_1$ | 23.47 | 22.48 | 20.07 | 22.80 | **17.37** |
| 250D | $W_2$ | 23.71 | 22.51 | 20.14 | 22.94 | **17.50** |
| | RBF-MMD | 0.55 | 0.18 | 0.12 | 0.16 | **0.11** |
| | $W_1$ | 29.46 | 28.88 | 22.60 | 24.97 | **20.99** |
| 500D | $W_2$ | 29.63 | 28.91 | 22.68 | 25.04 | **21.05** |
| | RBF-MMD | 0.51 | 0.15 | 0.12 | 0.13 | **0.12** |

Table 4: Comparison across methods on Single-Cell RNA dataset across embedding dimensions.

| | Metric | BSBM | CNF | CFM | FM | Y-Flows |
|---|---|---|---|---|---|---|
| | $W_1$ | 17.65 | 8.69 | 8.22 | 8.16 | **7.06** |
| 50D | $W_2$ | 18.11 | 9.14 | 8.52 | 8.41 | **7.28** |
| | RBF-MMD | 0.47 | 0.17 | 0.13 | 0.14 | **0.12** |
| | $W_1$ | 25.68 | 16.73 | 13.90 | 13.81 | **12.90** |
| 150D | $W_2$ | 26.13 | 17.13 | 14.23 | 14.17 | **12.68** |
| | RBF-MMD | 0.35 | 0.15 | 0.10 | 0.12 | **0.09** |
| | $W_1$ | 30.36 | 22.13 | 17.67 | 18.07 | **16.27** |
| 250D | $W_2$ | 30.77 | 22.49 | 17.95 | 18.36 | **16.46** |
| | RBF-MMD | 0.31 | 0.15 | 0.11 | 0.12 | **0.11** |
| | $W_1$ | 33.81 | 33.45 | 25.26 | 25.68 | **23.28** |
| 685D | $W_2$ | 34.07 | 33.56 | 25.45 | 25.88 | **23.44** |
| | RBF-MMD | 0.39 | 0.20 | 0.14 | 0.16 | **0.11** |

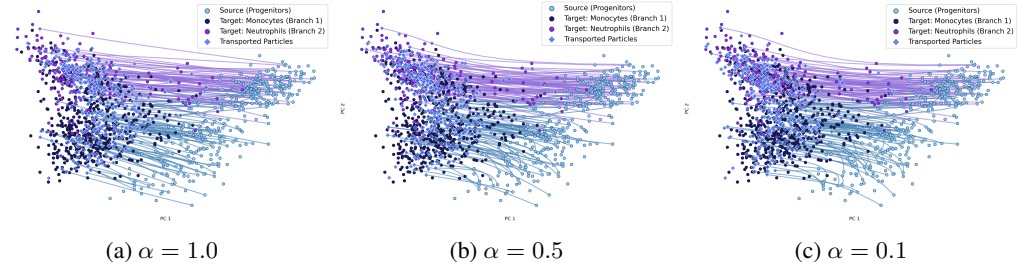

(a) $\alpha = 1.0$       (b) $\alpha = 0.5$       (c) $\alpha = 0.1$

Figure 7: Ablation on $\alpha$ for the Paul15 single-cell RNA dataset. As we can see, little branched structures appear as the value of $\alpha$ decreases. In this dataset, the source and target are located very close to each other in the Euclidean space, so, strong branching does not appear.

The implementation *Mean Flows* was based on:
Gsunshine/py-meanflow.

The experiment on *Domain Translation* was inspired by:
milenagazdieva/LightUnbalancedOptimalTransport.

FFHQ *encoded dataset and decoder* are taken from:
podgorskiy/ ALAE

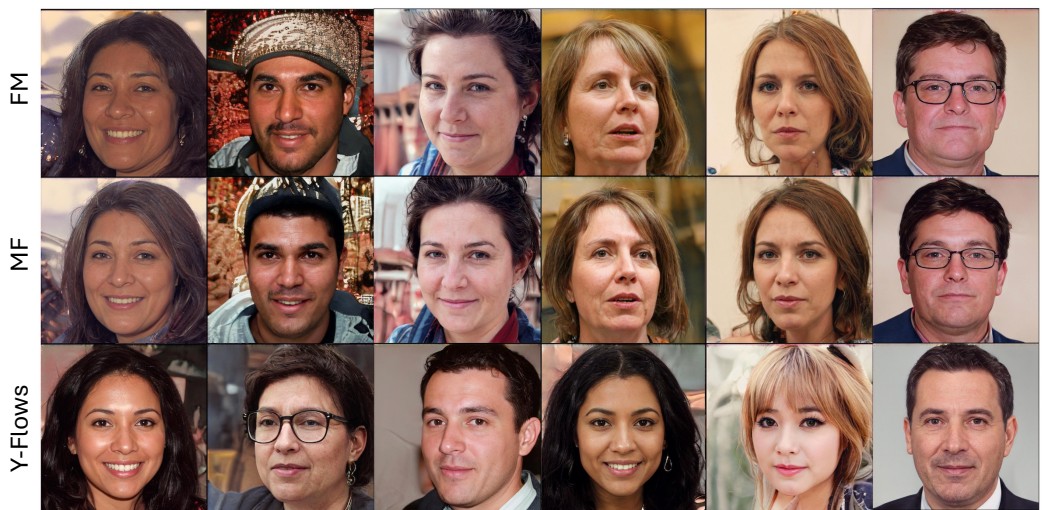

Figure 8: Image generation. Each row depicts a result of a generative model predicting a sample from the target distribution $\mu_1$. Samples from the source distribution $\mu_0$ are fixed and are the same for all models.

### F.4 FLOW-VISUALIZATION EXPERIMENTS

To qualitatively compare the geometric structure of the learned flows, we visualize the trajectories from random Gaussian noise to MNIST using a PCA projection (computed over the full set of $28\times28$ trajectories). Figure 9 contrasts Y-Flows with standard Flow Matching. In the projected space, Y-Flows exhibits a visible though slightly compressed Y-shaped geometry: trajectories initially move collectively along a shared trunk before separating. The slight attenuation of the branching angle is expected, since projecting a high-dimensional branched flow onto a 2D PCA subspace can collapse some of the orthogonal branching directions, causing the true Y-shape to appear narrower. Flow Matching, on the other hand, remains distinctly V-shaped under the same projection, with trajectories diverging immediately and independently, consistent with its objective of regressing straight source–target directions.

### F.5 USE OF LLM

We used an LLM for grammar editing.

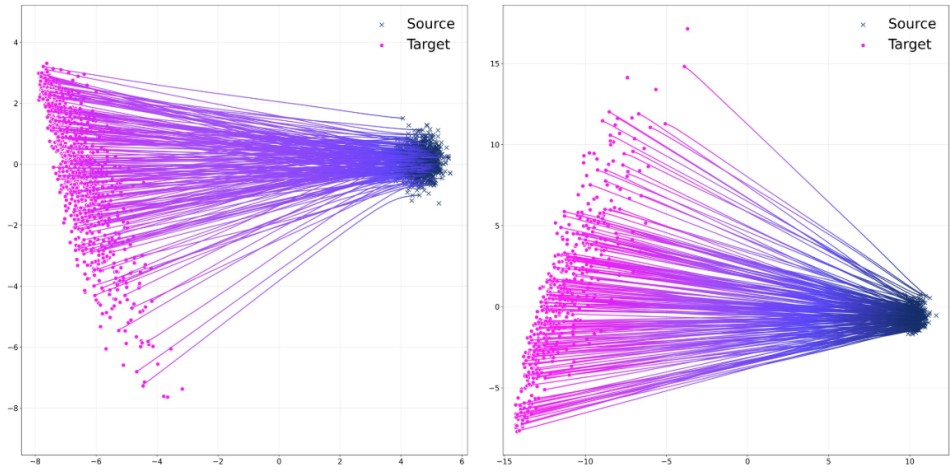

Figure 9: **Flow geometry on MNIST.** Left: Y-Flows. Right: Flow Matching. Each panel shows a 2D PCA projection of trajectories in the original $28 \times 28$ pixel space. Y-Flows exhibits a compressed but still recognizable Y-shaped structure due to projection, while Flow Matching remains strictly V-shaped with immediate, independent divergence of paths.

