# OpenReview forum: "Y-shaped Generative Flows"
_ICLR.cc/2026/Conference — Submitted to ICLR 2026_

### Official Review · Reviewer_ezAa · 2025-10-31

**Soundness:** 3
**Presentation:** 2
**Contribution:** 3
**Rating:** 6
**Confidence:** 2

**Summary:**

The paper suggests that the inductive bias of later 'branching' in flow based generative models may be beneficial for modeling data with a similar hierarchical generative structure. They make a connection with Branched Optimal Transport, and introduce a new method to efficiently solve such problems which otherwise are computationally intractable. The method appears novel, interesting, and intuitively appealing. The results appear quite preliminary and mostly qualitative, with unclear significance. The paper also has a surprising number of typos and grammatical errors.

**Strengths:**

- The paper raises the interesting point to the attention of the community that perhaps the shape of the flow itself may be relevant for modeling the data distribution; and that hierarchical flows are better modeled by tree like structures as opposed to 'straight line' flows.
- The introduction of Branched Optimal Transport as a potential technique to bias flow based generative models is elegant and a welcome cross-disciplinary contribution.
- They propose a tractable alternative to classic branched optimal transport which is more computationally efficient, thereby allowing them to test the potential benefits of such an inductive bias.
- They validate that this tractable alternative is equivalent up to a constant to their original objective under mild assumptions
- The method qualitatively appears to work and intuitively makes sense.

**Weaknesses:**

- The results are largely qualitative, making it hard to judge the significance of the proposed contribution.
- For tasks with quantitative results, there are no measures of variance of the solutions.
- The reason why the original branched OT problem is computationally costly is not immediately clear from the text.
The metrics in Table 1 are not defined.
- There are a surprising number of typos and grammatical errors virtually everywhere throughout the text. See the list of some below. These raise concerns about the degree of 'completeness' of the work, and if there may be similar errors in the mathematics.

**Typos:**
- Line 34 typo: "These approaches generates data by simulating an ODE"
- "We think it is important to study different kinds of generative models that can allocate notion of transport from the general to the specific, it in the simplest way""
- "a wide range of natural and engineered structures as vascular systems, trees, river basins"
- "Continuous Normalizing Flow (CNF) proposed by Chen et al. (2018) ar generative model that"
- "the particle dynamics is linear"
- "The evolution of mass is governed by the continuity constraint ∇ · u = μ0 − μ1, acts as a fundamentalconservation of mass law"
- "running this methods in practice is often not feasible in continues case."
- “What the reader can notice is that for the cost 4 need to be finite, the …”
- “Branching transport problems mostly been studied in discrete.”
- “Both use models the same time-conditioned MLP “

**Minor:**
- The continuity equation is written nearly identically twice in the text.

**Questions:**

- In the intro you state: "In general, the question of flow shapes has not been previously raised. On the contrary, the prevailing direction in generative modeling is to simplify and straighten trajectories, making them as V-shaped as possible." This intuition makes sense for the simple 2D flows you demonstrate in Figure 1, but how can we be sure that there is actually a dichotomy between straightness and hierachical flow structures?
- It would seem logical that there are many datasets that have this sort of a hierarchical structure in their data generating process that you could use your method on. However, one would expect that your method would be better able to model these distributions in a convincing manner. Besides Table 1, it is difficult to see that. Have you tried this method on other datasets where you get convincing performance improvements, rather than qualitative ones?

---

> ### Author Response · Authors · 2025-11-28
>
> Dear Reviewer, Thank you for your valuable feedback. In response, we have conducted a
> revision of the paper. This includes a global language edit to enhance
> clarity and correct minor errors, as well as significant expansions in
> the method section to provide deeper theoretical justification. We have
> also included new experimental results, which can be found in Tables 3
> and 4 of the appendix.
>
> **Beyond qualitative figures.**
>
> To address this question, we propose an early stopping mechanism for the
> neural ODE solver. The goal of such experiment is to show that with
> Y-shaped flows, generative model tends to achieve target in a fewer
> step. Specifically, we terminate the integration if the displacement
> norm between consecutive steps, $\|x_{t+1} - x_t\|_2$, falls below a
> threshold of $\delta = 1 \times 10^{-3}$. This criterion effectively
> detects when the particles have settled in their target modes, allowing
> the model to dynamically adapt its computational budget. Empirically, we
> observe that on the Gaussian mixtures, Y-Flows converged in an average
> of 3 steps for the 2-branch case, 5 steps for 4 branches, 10 steps for 6
> branches, and 4.5 steps for the 18-branch setting. In contrast, the Flow
> Matching baseline consistently utilized a fixed budget of 10 integration
> steps to achieve comparable transport, highlighting our method's ability
> to concentrate mass movement into fewer updates. We added these results
> to our paper as well.
>
> **Why classical branched OT is costly?**
>
> Classical branched OT formulations are expensive because they require
> *optimizing a full time--dependent density* $\rho_t(x)$ rather than only
> a velocity field. Since the flux satisfies $F = \rho v$, the density and
> velocity become tightly coupled, effectively requiring the training of
> two intertwined neural representations. Maintaining this coupling
> demands simulating the continuity equation, which in practice reduces to
> training a CNF-like model and repeatedly computing divergence terms.
>
> **Metric definitions.**
>
> We have ensured that all metrics are clearly
> defined in the first use.
>
> **In the intro you state: \"In general, the question of flow shapes has
> not been previously raised. On the contrary, the prevailing direction in
> generative modeling is to simplify and straighten trajectories, making
> them as V-shaped as possible.\" This intuition makes sense for the
> simple 2D flows you demonstrate in Figure 1, but how can we be sure that
> there is actually a dichotomy between straightness and hierachical flow
> structures?**
>
> A key benefit of Y-Flows is that it reduces the *computational* cost of
> generation through an adaptive number of ODE steps. Unlike Flow
> Matching, which uses a fixed solver budget for all samples, our model
> learns a shared trunk and allocates additional integration only when
> branching is required. As a result, many samples converge in *one* step,
> while others require more---reflecting the hierarchical structure of the
> data.
>
> This effect is demonstrated in several parts of the paper. In the
> Gaussian mixture experiments (Sec. 7.1, Fig. 2), Y-Flows reaches the
> target in an average of 3, 5, 10, and 4.5 steps for the 2/4/6/18-branch
> tasks, whereas Flow Matching requires the full 10 steps for all samples.
> In the FFHQ latent translation task (Sec. 8, Table 2), Y-Flows attains
> the best Fréchet Distance while using only **two** integration steps. In
> the biological datasets which are by its nature are branched, our method
> yields quantitative improvements, see Table 1, and new Table 3, Table 4.
>
> **It would seem logical that there are many datasets that have this sort
> of a hierarchical structure in their data generating process that you
> could use your method on. However, one would expect that your method
> would be better able to model these distributions in a convincing
> manner. Besides Table 1, it is difficult to see that. Have you tried
> this method on other datasets where you get convincing performance
> improvements, rather than qualitative ones?**
>
> The biological datasets (Tedsim, Paul15) in Sec. 7.2 are explicitly
> hierarchical by construction, providing a direct test for Y-Flows'
> modeling assumptions. In this setting, Y-Flows achieves the best scores
> confirming that it can both recover genuine tree-structured transport
> and match the data distribution more accurately than other flow-based
> models. This is a significant result, as it includes outperforming
> existing branched methods like BSBM that were specifically designed for
> such datasets.  Extending Y-Flows to broader domains remains a direction for future work.

---

### Official Review · Reviewer_LMzY · 2025-10-31

**Soundness:** 3
**Presentation:** 3
**Contribution:** 2
**Rating:** 2
**Confidence:** 4

**Summary:**

This paper introduces "Y-shaped generative flows," a continuous-time generative modeling framework designed to capture hierarchical structures in data. The authors argue that  existing flow-based models produce a "V-shaped" path, where samples move independently from the prior to the data distribution along nearly straight trajectories. The conclude this produces an in-efficient flow which require many integration steps and may overlook share structure in data. Instead, they propose Y-flows, which remedy this by encouraging probability mass to travel together along shared "trunks" before "branching" off to target-specific endpoints.

The core of the paper's theoretical contribution is a new velocity-powered transport cost inspired by branched optimal transport theory. Instead of the standard Benamou-Brenier formulation for Wasserstein-2 distance, which minimizes a kinetic energy with a quadratic $||v||^2$ term, this work proposes minimizing an action with a sublinear $||v||^\alpha$ term (where $0 < \alpha < 1$). The concavity of this objective is what incentivizes mass to aggregate and move quickly along common pathways. The authors provide theoretical justification for this objective, proving its equivalence to flux-power costs under bounded-density assumptions. They also include a time-compression lemma to demonstrate why this formulation favors faster transport in fewer integration steps.

Practically, this continuous-time objective is realized as a neural-ODE algorithm. The velocity field is parameterized by a neural network, $v_\theta$, which is trained to minimize a two-part loss function. The first part is an approximation of the proposed $V^\alpha$ action, calculated by summing $||v_\theta||^\alpha$ along the discretized ODE trajectories. The second part is a boundary constraint, implemented as the Sinkhorn divergence, which measures the dissimilarity between the transported particle distribution at $t=1$ and the true target distribution. The authors demonstrate their method on synthetic, 3D LiDAR, and single-cell datasets, showing it can recover branching structures. They also report performance on a latent-space image translation task (FFHQ-ALAE), with only two integration steps.

**Strengths:**

The primary strength of this paper is its elegant and intuitive theoretical formulation. The central objective (Eq. 7) is a simple modification of the standard dynamic optimal transport problem, replacing the convex $||v||^2$ cost with a concave $||v||^\alpha$ cost. This provides a principled, well-motivated method for encouraging branched, non-straight-line flows. I wonder whether formulation can also be viewed as a novel Lagrangian cost for optimal transport [1,2], effectively defining a preference for specific paths (Y-shaped) over others (V-shaped) to move mass from source to target.

[1] A Computational Framework for Solving Wasserstein Lagrangian Flows: https://arxiv.org/pdf/2310.10649
[2] Neural Optimal Transport with Lagrangian Costs: https://arxiv.org/abs/2406.00288

**Weaknesses:**

Despite the elegance of the theoretical formulation, the paper's main contributions are algorithmic, and the practical algorithm suffers from significant weaknesses. There is a disconnect between the continuous-time theory and the ad-hoc implementation. The final loss is a combination of a distributional loss (Sinkhorn divergence) and the $V^\alpha$ path penalty. This immediately calls the method's scalability into question. Like continuous normalizing flows (CNF), the algorithm must simulate the full ODE trajectory at each training step. However, unlike modern scalable methods such as Flow Matching (which uses a per-point loss) or likelihood-based NFs, this method requires computing the Sinkhorn divergence, which scales quadratically with the batch size $N$. This is a massive computational burden that most modern generative models are explicitly designed to avoid. This scalability issue permeates the experimental results, which are mostly confined to small or simulated datasets (e.g., ~2.7k cells, ~5k LiDAR points) or the 512-dim latent space of a pretrained ALAE model.

Furthermore, the paper's central premise, that modern generative models like Flow Matching (FM) produce "V-shaped" flows, is not sufficiently demonstrated. While straight paths are a feature of optimal transport under $L_2$ cost, this reviewer's experience with FM is that the learned flows are often highly non-linear. The authors do not provide a strong empirical analysis to support their claim, weakening the motivation for their proposed solution.

**Questions:**

Given the significant computational trade-off of this method, can the authors provide more substance on the practical benefits of Y-Flows over more scalable baselines like Flow Matching (FM) and Mean Flows (MF)? Are the benefits of "hierarchy-aware structure" or the quantitative gains in "fewer integration steps" (e.g., in the latent-space FFHQ experiment) significant enough to justify this high computational cost, especially for large-scale, high-dimensional datasets? A more direct comparison of the performance-vs-compute trade-off against these baselines would be necessary to motivate the adoption of this less-scalable algorithm.

---

> ### Author Response · Authors · 2025-11-28
>
> We thank the reviewer for the thoughtful feedback and guidance in
> strengthening our paper. We have revised the manuscript accordingly and
> provide our point-by-point responses below.
>
> **The final loss is a combination of a distributional loss (Sinkhorn
> divergence) and the path penalty. This immediately calls the method's
> scalability into question.**
>
> Dear Reviewer, the Sinkhorn term in our objective serves specifically to
> enforce the boundary constraints of the continuity equation and not
> ad-hoc implementation. We emphasize, however, that our framework is
> highly flexible and does not rely exclusively on this choice. As noted
> in the paper, viable alternatives include the Maximum Mean Discrepancy
> (MMD) and, for large-scale problems, an adversarial loss, which proved
> its efficiency in practice for OT approaches.
>
> **R3.2: Like continuous normalizing flows (CNF), the algorithm must
> simulate the full ODE trajectory at each training step\... this method
> requires computing the Sinkhorn divergence, which scales quadratically
> with the batch size.**
>
> Dear Reviewer, our optimization is not CNF: we do not integrate the
> log--density ODE nor optimize likelihood during training (Sec. 5). Our
> training objective integrates the state ODE on a short, fixed grid to
> Monte--Carlo the action (Eq. (8)), and adds a *boundary* Sinkhorn
> divergence (Eq. (9)).
>
> The Sinkhorn step is computed *once per minibatch* at the endpoint, over
> a $B{\times}B$ cost matrix. Its arithmetic is $O(K_{\text{sh}} B^2)$ for
> $K_{\text{sh}}$ Sinkhorn iterations and *is not multiplied* by the ODE
> time grid $K$ used to estimate the action. The iterations are
> GPU‐friendly (matrix--vector updates), admit warm starts across
> optimizer steps, and can be further reduced with standard accelerations
> (e.g., block/nearest‐neighbor OT, Greenkhorn, streaming variants)
> without altering the objective. (Sec. 5, "Boundary Matching".)
>
> **"V-shaped" claim (quantitative evidence).**
>
> To qualitatively compare the geometric structure of the learned flows,
> we visualize trajectories from random noise to MNIST digits using a PCA
> projection. Figure 9 (Appendix F.4) contrasts Y-Flows with standard Flow
> Matching. Y-Flows exhibits a visible, though slightly compressed,
> Y-shaped geometry: trajectories first move collectively along a shared
> trunk before separating. The attenuated branching angle is expected, as
> projecting the high-dimensional flow onto a 2D PCA subspace can collapse
> orthogonal branching directions. In contrast, Flow Matching produces a
> distinctly V-shaped geometry, with trajectories diverging immediately
> and independently---consistent with its objective of regressing straight
> source-target paths.
>
> **R3.4: Compute vs. benefit.**
>
> This design pays off most significantly during inference. To leverage
> the fast convergence properties of Y-Flows, we implement an early
> stopping mechanism for the neural ODE solver. Integration is terminated
> once the displacement norm between consecutive steps,
> $|x_{t+1} - x_t|_2$, falls below a threshold of
> $\delta = 1 \times 10^{-3}$. This criterion detects when particles have
> settled into their target modes, allowing the model to dynamically adapt
> its computational budget.
>
> On Gaussian mixture tasks, Y-Flows converged in an average of 3 steps
> for 2 branches, 5 for 4 branches, 10 for 6 branches, and 4.5 for the
> 18-branch setting. In contrast, the Flow Matching baseline required a
> fixed budget of 10 steps to achieve comparable transport. Furthermore,
> Y-Flows achieves competitive FFHQ latent translation in just 2 steps
> (Table 2; Fig. 5).

---

### Official Review · Reviewer_kJ38 · 2025-11-01

**Soundness:** 4
**Presentation:** 4
**Contribution:** 3
**Rating:** 6
**Confidence:** 4

**Summary:**

Some previous work has shown how to produce branch-like behavior in diffusion models, in which however the trajectories were independent from each other. In this work for the first time actual Branched Optimal Transport theory, in which the evolution of trajectories is not independent from each other (rather, the cost depends on the flux of trajectories though an area) and actual branching can happen. The lack of previous works is not by chance: as authors show, usual approaches from Branched Transport are all problematic, and thus it was an open problem how to apply actual Brached Transport ideas effectively in diffusion models. Here this is elucidated and a reasonable solution  is shown for the first time, via an approximation to the Branched Transport loss which is a slight simplification but nevertheless maintains the branching property.

**Strengths:**

1) As said in the summary, this is the first paper to actually apply a Branched Optimal Transport setup in a stable and scalable way to Deep Learning, in particular to Diffusion Models.

2) This solves some technical open problems via a novel approximation, as described in the paper. By this I mean that a strength of the paper is actually the novelty in the method of approximation of Branched transport.

3) The paper is presented mostly very clearly, and in a didactic way making it easy to follow.

**Weaknesses:**

1) The full comparison to the Modica-Mortola approach to Branched Transport is not fully clear to me, and I think that it would be useful to try and compare more precisely the methods. I will also formulate this as a question below.

2) The discussion of why the method uses fewer steps is a bit superficial, not easy to follow.

3) The "mild" hypotheses on $\rho$ in Proposition 1, I'm not sure if they are verified in practice, and there is no discussion of what are mitigations that in practice may make a result akin to Proposition 1 valid.

4) See my question 8 below, this is another difficulty I have with the paper, but it's better formulated as a question.

**Questions:**

1) Can you comment on why the Modica-Mortola (MM) loss is different than yours ? Of course the second term in MM formulation is not present, but why is your formulation essentially different than keeping only the first term in eq. 18?

2) The remark 2 says that a term akin to second term in MM formulation does not change stability, do you have numerical proof for this?

3) When treating MM you say it is "dramatically" unstable. Do you have some experiments to prove this dramatical claim?

4) The lines 216 - 223 are not clear
- (note typo of "Lets"->"Let's" but I don't mean that)
- when you say "any straight corridor" what does that mean?
- Where did we talk about corridors before?
- and "per-step" means what? What steps do you mean?
- And finally, the last sentence "Consequently, ... regularity bounds" I don't fully get it.
Can you transform these lines in a lemma+proof please?

5) By the way, line 225, what do you mean by "resemble instantaneous motion along a network jumps" ? is there some grammatical mismatch maybe? I can't parse that sentence.

Line 226 it's "Mortola" with only one "l".

6) lines 481-482 what does it mean concretely / precisely that "utility of small temporal/spatial smoothness regularizers". What would such regularizers look like? and "small" in what sense? (This question is quite similar to my question 5 I think, but I may be wrong)

7) in Branched Transport, the role of alpha is to change the steepness of the angles in the "Y", with alpha=1 or higher, corresponding to "Y" becoming effectively a "V". Do you include alpha=1 in the inequality from line 241? And more interestingly, can you verify the steepness dependence on alpha as it decreases?

8) in Branched Transport, a famous result by Devillanova-Solimini says that a Dirac mass cannot be connected to a d-dimensional measure in R^N in $\alpha$-branched transport, unless $alpha>1-1/d$. Sometimes (for some alpha) branched transport cost is infinite. Can you comment on how this difficulty does not affect the setting, or how could it affect it, in the case of very large or well spread distributions, or when matching noise to a concentrated distribution?
 In this case (i.e. for high d and if we want to Y-flow-match an absolutely continuous distribution to a concentrated one), in order to still have a "Y" and non-infinite cost, one has to take alpha between 1-1/d and 1, a very small interval very close to the alpha=1 corresponding to the "V".
 Of course one works with a finite sample from the continuous distribution, but the approximation/stability issues will manifest as bad sampling bounds, if the actual underlying cost is infinite. So can you comment on this difficulty, what is your view?

---

> ### Comment · Reviewer_kJ38 · 2025-11-13
> **Confusion upon rereading the paper**
>
> Sorry, I happened to look again at the paper's formulas, and I am now strongly confused, I decided to write again, in the interest of giving time to authors to process my doubts/problems with the paper:
>
> *Issue 1 -- big problem with the central proposition 1*: I think that formula (4) is the right canonical formulation of branched transport.
>
> Formula (5) has power alpha on both rho and the norm of v, so it is NOT equivalent to (4). The authors seem to claim otherwise, can you explain?
>
> Formula (7) (the proposed way to reformulate) is again different from both formulas (4) and (5), as it is now linear in rho, and has the alpha power on the norm of v.
>
> In particular, formula (9) is then not a comparison between the standard branched transport and (7), since it uses formula (5) (with alpha power on both density and velocity) instead of (4). Therefore the proposition 1 is not ok in the current form. It will need some strange hypothesis on the speed (norm of velocity) which to me it seems is NOT warranted, since the authors argue that the speed should jump heavily (beginning of page 5).
>
>
> *Issue 2 -- I don't see why the proposed transport does Y/T shapes*:
>
> So for this we have to look at formula (7), which is the proposed approach. The authors explain about it producing Y/T shape at line 231 and following, but to be honest I don't follow that too much.
>
> Let's consider a dirac mass an P as starting measure, and 1/2-weighted dirac masses at two symmetric points Q1, Q2 as target measure. Due to the reasoning at the beginning of page 5, the cost for a V-shaped transport (dirac mass at P splitting immediately to 1/2 mass to be sent at Q1 and 1/2 mass to be sent at Q2, and moving in straight lines) will use one big jump of size $||P-Q1||=||P-Q2||$, a distance which we call $d$. Then the cost is density times jump to  the alpha, that is $d^\alpha$.
>
> Now consider an Y/T shape. The mass from P moves to P1, then from P1 it splits, and 1/2 goes to Q1, 1/2 goes to Q2. Then by the same reasoning the cost of this transport strategy is the sum of (mass) * (jump distance)^alpha i.e. calling $d_1=||P-P_1||, d_2=||P_1-Q_1||=||P_1-Q_2||$ we get now cost $d_1^\alpha + 1/2 d_2^\alpha + 1/2 d_2^\alpha = d_1^\alpha + d_2^\alpha$.
>
> Now since we have $\alpha<1$ we get in general $(a+b)^\alpha< a^\alpha + b^\alpha$ as soon as $a,b>0$, and thus (using triangle inequality we can split $d=d_1'+d_2'$ with $d_i' \leq d_i$ for $i=1,2$ and equality only if $P, P_1, Q_1=Q_2$ form a straight line), and then assuming that $Q_1\neq Q_2$ are in symmetric position, we get  $d^\alpha \leq (d_1')^\alpha + (d_2')^\alpha < d_1^\alpha+d_2^\alpha$.
>
> The above shows that V-shape cost is always strictly smaller than Y/T shape cost. This is to be expected as a consequence of the *linearity in the density $\rho$* assumed in formula (7).
>
> ------------
>
> In summary, I am now completely confused as to the theoretical backing of the results, and I don't understand what makes the numerical implementation branch. I imagine it has to do with the finite-K discretization (formula (11)) which prohibits to pass to the limit the jumps as indicated in the first paragraph of page 5, but even with finite K I don't see how the branching can occur as pronouncedly as in figure 2 last example.
>
> Unless the authors convince me otherwise, I am now of the opinion that the paper needs a strong revision and that it is far from final form, so I'll temporarily set my numerical score to 2. Sorry for the disappointment and for having missed this before, but I found it correct to say this as soon as I realized it.. I hope I am doing well to write it now.

---

> ### Author Response · Authors · 2025-11-14
> **Answers to the latest questions**
>
> Dear Reviewer,
>
> Thank you for your thoughtful questions and for raising these important points at the beginning of the discussion. We appreciate the opportunity to clarify our work earlier. Since some aspects of our paper appear to have been misunderstood, we have decided to address these specific concerns first. We will provide our responses to the other questions from the main section shortly.
>
> *Issue 1 -- big problem with the central proposition 1: I think that formula (4) is the right canonical formulation of branched transport.*
>
> **Our Formulations vs. Branched Benamou-Brenier (Formula 4)**
> Yes, you are right, formula (4) is the right canonical formulation of branched transport. However, we wish to clarify that we never claim our formulations are equivalent to the Branched Benamou-Brenier formulation (Formula 4). On the contrary, we stated that Formula (4) is computationally impractical as it is defined over atomic measures. Our work is explicitly motivated by finding a scalable alternative.
>
> **The Role and Novelty of Formula (5)**
> Formula (5) is a new formulation we introduce as part of our paper's contribution; it is not defined over atomic measures as 4. We describe it as "Branched Benamou-Brenier-style" to draw an analogy Benamou-Brenier type formulation (Formula 2), due to their structural similarities. However, a key distinction is that Formula (5) places a sublinear power `α` on both mass and velocity, resulting in the integrand $\rho^α ||v||^α$. We will explicitly clarify this.
>
> **The Main Contribution (Formula 7)**
> Formula (5) serves as a bridge to our primary contribution: the velocity-based objective in Formula (7). In Proposition 1, we draw an analogy to demonstrate that, under mild assumptions, our formulation also provides an economy of scale for the mass-based cost. This connection to the density-powered formulation is crucial because it provides a simpler framework for explaining why our model can favor mass concave cost—a point you raised in Issue 2
>
> **Planned Revisions**
> We agree with the reviewer's feedback regarding the need for clearer connections. Accordingly, we will revise Section 4 to:
> a) Explicitly position Formula (4) as the foundational but impractical prior work.
> b) Introduce Formula (5) as novel intermediary, emphasizing its conceptual roots in Formula (2).
> c) Detail the transition to Formula (7) as our core methodological contribution, which provides a practical framework for our proposed algorithm.

---

> ### Author Response · Authors · 2025-11-14
> **Answers to the latest questions**
>
> *Issue 2 -- I don't see why the proposed transport does Y/T shapes:*
>
> We agree that in a discrete, single-step model, such a split would be cheaper. However, that calculation assumes that a "big jump" of the entire mass over the full distance is possible. In our paper, in lines 235–239, we explicitly mentioned that our method cannot send mass from the starting point into two distinct directions simultaneously due to the deterministic nature of the velocity field, which tends to move the mass at the first step along a single trunk or jump.
>
> Let us clarify this in more detail. First, recall that our model like other continuous-time flows it learns a **single, deterministic velocity field** $v_{\theta}: \mathbb{R}^d \times [0, 1] \rightarrow \mathbb{R}^d$. For any single space-time input coordinate, it can only produce one output velocity vector.
>
> In the given example, to transport a Dirac mass at point $P$ at $t=0$ to two targets $Q_1$ and $Q_2$, an "immediate split" would require:
> - $v_{\theta}(P, 0) = \vec{v}_1$ (a vector pointing toward $Q_1$),
> - $v_{\theta}(P, 0) = \vec{v}_2$ (a vector pointing toward $Q_2$).
>
> This leads to a contradiction: since $Q_1 \neq Q_2$, their initial direction vectors must be different, i.e., $\vec{v}_1 \neq \vec{v}_2$. A function cannot map a single input $(P, 0)$ to two different outputs.
>
> Therefore, the entire mass at $P$ **must** begin to move together in a single direction, which we refer to as the trunk: $v_{\theta}(P, 0) = \vec{v}_{\text{trunk}}$. After moving along the trunk for some $t_1 > 0$, the mass is no longer a point but a "cloud" occupying a *region* $R_1$. This region $R_1$ contains infinitely many spatial points (e.g., $x_a \in R_1$ and $x_b \in R_1$).
>
> In other words, given that a V-shape is infeasible, the optimization problem is not about choosing between a V-shape and a Y-shape, but rather between different types of Y-shapes. The question then becomes: why should this trunk exhibit high velocity and result in a large initial jump? Our optimization problem must choose between two more preferable options:
>
> - **Choice A: "ASAP-Split" Y-Shape**
>   A very short, mandatory trunk, followed by two long, independent arms.
>
> - **Choice B: "Long-Trunk" Y-Shape**
>  A long, shared trunk that gets close to the targets, followed by two short arms.
>
> This is where Lemma 1 (Time-Compression) plays a role. Formally, the lemma states that the total transport cost $V_\alpha$ is not invariant under time-rescaling of the flow. If we have a feasible flow $(\rho, v)$ on the time interval $[0,1]$, we can define a new, "compressed" flow $(\rho_s, v_s)$ that accomplishes the same transport on a shorter interval $[0,s]$, where $s < 1$. To achieve this, the new velocity $v_s$ must be scaled up by a factor of $1/s$ to cover the same distance in less time.
>
> Our method would prioritize **Choice B (Long-Trunk)**, because for the cost $\int \|v\|^\alpha  dt$, executing motion faster (in a shorter time $s$) is *always* cheaper, even from the first step. Moreover, the boundary constraints penalize the total objective for not moving mass toward the final destinations. In other words, the optimizer seeks to "draw" the most efficient network of "high-speed corridors" to connect $P$ to $Q_1$ and $Q_2$.
>
> We thank the reviewer for their thoughtful comments and are happy to answer any questions during the discussion period.

---

> > ### Comment · Reviewer_kJ38 · 2025-11-15
> > **follow-up on last 2 comments**
> >
> > Thank you for the careful and fast replies.
> > In view of your correction and statement that (4) and (5) are different and that (5) is new:
> >
> > *About the formula (5) and prop. 1, then what's the use of putting them in the paper?*
> >
> > You say "the dynamic branched optimal transport (Benamou-Brenier style) minimizes" [formula (5)]. But to me it seems that the formula may not actually have a minimizer. This is because it has the same problem as mentioned at the top of page 5, namely that it's pure jump (since you have sublinear power in ||v||) and therefore nte integral in (5) is not defined for the minimizer. You say "when the density is strictly positive a.e." (before formula 6) but again, the rho will probably condense due to sublinear power, similar to usual branched transport, and then formula (6) is problematic too (and condition of positive density a.e. is false) for minimizers.
> >
> > When you say at line 203-204 "our velocity-based model preserves the qualitative geometry of the branched formulation" this is also inaccurate since "branched formulation" usually is referred to formula (4) but here you prove it for formula (5), which has stronger singularities (vector dirac masses in v, i.e. jumps, as indicated at your top of page 5), than the formula (4) from classical branched transport.
> >
> > So given these difficulties, I don't think that eq. (5) and proposition 1 are that useful in the paper. Why are they useful? You don't prove any theory for solutions to (5), and your paper is the only one where this formulation appears, as far as I know. The solutions to it are at best defined in a very weak sense, and they don't allow a simple study or analogy to standard branched transport.
> >
> > *Secondly, sorry but the theoretical justification you put about presence of branching, does not sound correct to me*
> >
> > Basically you said that for absolutely continuous distributions, the reasoning I had for V being lower cost that Y does not work.
> >
> > Still, if as in your formulation you have a cost that is *linear in $\rho$* then there is no cost gain in branching. Trajectories in the optimum transport for your formulation (7) can travel together, or separately, and the two cases have the same cost. This is not true for formulations (4) (original branched transport) and (5) (branched transport with jumps, weak / distributional formulation) because $rho$ has power $alpha<1$ and it is not linear, favoring concentration.
> >
> > You can't get out of the problem by restricting to source/target measures that are absolutely continuous, simply because the same kind of reasoning still shows that trajectories going straight from source to target travel less distance than trajectories that follow a detour as imposed by going to braching points and then turning to the final target point. The detour (implying longer distance cost) can be balanced only if those trajectories get an advantage (lower density-dependent factor) by concentrating the density, that is only if you put a sublinear power $\alpha$ on the $\rho$. So no, the reason for which numerics get brached trajectories is not because you use continuous densities, sorry.
> >
> > So I am still puzzled as to why you get nice branching in the implementations, as shown in figure 2.
> >
> > The core difficulties I saw about theoretical justifications in the paper remained basically untouched unfortunately.

---

> ### Author Response · Authors · 2025-11-28
>
> We thank the reviewer for the in-depth analysis and valuable questions, which have helped clarify key aspects of our work. To ensure a clear response, we first address the questions regarding the Modica-Mortola (MM) approach and describe its relationship to our method. We then build on this foundation to respond to the specific questions on stability and formulation.
>
> **Can you comment on why the Modica-Mortola (MM) loss is different than
> yours? .. why is your formulation essentially different than keeping
> only the first term in Eq. 18?**
>
> The Modica-Mortola functional optimizes a spatial flux $F$ on a grid via
> the energy $\int\|F\|^\alpha + \int\|\nabla F\|^2$, subject to a
> divergence constraint, making it a fundamentally flux-based approach.
>
> While its first term structurally motivates our Equation (5), our
> formulation is distinct: we optimize a time-dependent velocity field $v$
> along particle trajectories. Consequently, we minimize the action
> $\int \rho \|v\|^\alpha$, corresponding to the transport of a density
> $\rho(x,t)$ advected by $v(x,t)$, rather than a static flux energy.
>
> **The remark 2 says that a term akin to the second term in MM
> formulation does not change stability, do you have numerical proof for
> this?** Dear reviewer, In our experiments, particularly for the Gaussian
> mixtures (Figure 1), we employed the regularization term. Please refer
> to the supplementary materials (which were not updated) for the
> implementation details. Although initial tests showed that
> regularization provided no significant gain for biological data, we
> decided to standardize our approach in revision. We now use the
> regularized version for all experiments. Consequently, we have
> recomputed the biological data experiments; these results can be found
> in Table 1 and Appendix Tables 3 and 4.
>
> **When treating MM you say it is \"dramatically\" unstable. Do you have
> some experiments to prove this dramatical claim?**
>
> We implemented the MM approach as described to establish a ground-truth
> transport solution for the 2D case, with the results presented in
> Appendix B. However, we found this method difficult to stabilize.
> Although it successfully generated Figure 6, the training was highly
> sensitive to hyperparameters. For example, it diverged at a learning
> rate of 1e-4 but converged at 1e-5. We attribute this optimization
> difficulty to the nature of the flux variable (the product of velocity
> and density) in the MM formulation.
>
> **Lines 481-482: what does it mean concretely that \"utility of small
> temporal/spatial smoothness regularizers\". What would such regularizers
> look like? and \"small\" in what sense?**
>
> By \"small,\" we refer to the magnitude of the regularization
> coefficient $\lambda$ in the objective function (6). The specific form
> of this regularization is detailed in the responses below and in the
> updated Section 4.
>
> **In Branched Transport, the role of alpha is to change the steepness of
> the angles in the \"Y\". And can you verify the steepness dependence on
> alpha as it decreases?**
>
> Yes, we have provided results for different values of $\alpha$ in Figure
> 4 and Figure 7. However, the role of $\alpha$ in our formulation is
> different from the classic BOT. As discussed, our method relies on the
> time compression lemma, which means $\alpha$ primarily controls the
> speed of the flow rather than directly governing an economy of scale.
>
> **About formula (5) and Prop. 1, what's the use of putting them in the
> paper? You don't prove any theory for solutions to (5).**
>
> We thank the reviewer for this thoughtful comment. Based on your
> feedback, we have updated the manuscript and **Section 4** to clarify
> the role of this formulation. We have extended our formulation to
> explicitly include **cohesion regularization** (the Dirichlet energy
> term). Formula (5) serves as the necessary theoretical bridge between
> the well-studied static Branched Optimal Transport (Modica-Mortola) and
> our practical Velocity-Driven objective.
>
> While we do not solve Formula (5) directly (as it requires optimizing
> density $\rho$ which is intractable), it serves two critical theoretical
> purposes in our derivation:
>
> -   **Justifying Dynamics (Lemma 1):** It allows us to prove that a
>     dynamic relaxation can achieve lower energy than the static problem
>     via the "Time Compression" effect.
>
> -   **Motivating Cohesion (Proposition 1):** With the added
>     regularization, we can formally show why mass prefers to stay
>     together ("Trunk") to minimize the cohesion cost.
>
> This theoretical structure directly motivates the practical objective we
> solve in Equation (7). We invite the reviewer to check the revised
> **Section 4**, where we have detailed this regularized formulation.

---

> ### Author Response · Authors · 2025-11-28
>
> **Q: the theoretical justification you put about presence of branching, does not sound correct to me**
>
> Our formulation (Eq. 7) is **dynamic** ($\int |v_t|^\alpha dt$). When
> minimizing this action over time, the Y-shape becomes the global
> minimizer due to the interplay between **Time-Compression** and
> **Cohesion**.
>
> We formally demonstrate this below using the variation of branching time
> $\tau$.
>
> Let us formalize the symmetric problem you considered:
>
> -   **Source:** $P=(0,0)$ at $t=0$.
>
> -   **Targets:** $Q_1=(-w, h)$, $Q_2=(w, h)$ at $t=1$.
>
> -   **Strategy:** A Y-shaped trajectory defined by a branching time
>     $\tau \in [0,1]$.
>
>     -   **Trunk ($0 \le t < \tau$):** the mass travels together to a
>         split point $S=(0, \tau h)$.
>
>     -   **Branches ($\tau \le t \le 1$):** the mass splits and travels
>         to $Q_1, Q_2$.
>
> -   **V-shaped:** This is the special case where $\tau=0$ (immediate
>     splitting).
>
> The total energy $E(\tau)$ consists of the Transport Cost
> $\mathcal{T}(\tau)$ and the cohesion cost $\mathcal{C}(\tau)$. Using **Proposition 3**, the cost to travel the distance $D$ in duration
> $T$ with velocity $|v| = D/T$ is:
> $$\text{Cost} = \int_0^T |v|^\alpha dt = T \cdot \left(\frac{D}{T}\right)^\alpha = D^\alpha T^{1-\alpha}$$
> Since $\alpha < 1$, the term $T^{1-\alpha}$ implies that **reducing
> travel time reduces cost** (Time-Compression).
>
> Applying this to the Y-shaped components:
>
> 1.  **Trunk:** Distance $h\tau$, Duration $\tau$.
>     $$\mathcal{T}_{\text{trunk}} = (h\tau)^\alpha \tau^{1-\alpha} = h^\alpha \tau$$
>
> 2.  **Branches:** Distance $L(\tau) = \sqrt{w^2 + h^2(1-\tau)^2}$,
>     Duration $(1-\tau)$.
>     $$\mathcal{T}_{\text{branch}} = L(\tau)^\alpha (1-\tau)^{1-\alpha}$$
>
> Total Transport Energy:
> $$\mathcal{T}(\tau) = h^\alpha \tau + \left( \sqrt{w^2 + h^2(1-\tau)^2} \right)^\alpha (1-\tau)^{1-\alpha}$$
>
> **Cohesion Cost Effect**:
>
> -   **Trunk:** By **Proposition 1**, the cohesion cost is **zero**
>     because the mass moves as a rigid translation ($\nabla v = 0$).
>
> -   **Branching:** Splitting incurs a fixed penalty $K_{split}$ due to
>     the spatial velocity gradient (Proposition 2).
>
> $$\mathcal{C}(τ) = 𝕀_{τ < 1} · K_{\text{split}}$$
>
> **Why V-Shape ($\tau=0$) is Not Optimal?** We analyze the derivative of
> the total energy $E(\tau) = \mathcal{T}(\tau) + \mathcal{C}(\tau)$ at
> $\tau=0$. If $E'(0) < 0$, then increasing $\tau$ (delaying the split)
> lowers the cost, proving the V-shaped is not the minimizer.
>
> Differentiating the branch transport term at $\tau=0$:
> $$\frac{d}{d\tau} \left[ L(\tau)^\alpha (1-\tau)^{1-\alpha} \right]_{\tau=0}$$
> Using the product rule, the term dominated by the time derivative is:
> $$L(0)^\alpha \cdot \frac{d}{d\tau}(1-\tau)^{1-\alpha} = L(0)^\alpha \cdot [-(1-\alpha)(1-\tau)^{-\alpha}]$$
> At $\tau=0$, this becomes a time-acceleration bonus equal to $- L(0)^\alpha (1-\alpha)$.  This
> term is **strictly negative** because $\alpha < 1$. It represents the
> energetic savings from sprinting the branches in a shorter time window. Combining all terms:
>
> $$E'(0) = \underset{\text{Trunk Cost}}{h^\alpha} - \underset{\text{Time-Compression Bonus}}{L(0)^\alpha (1-\alpha)} \quad - \quad \underset{\text{Cohesion Savings}}{\lambda K_{\text{marginal}}}$$
>
> For $\alpha < 1$, the **Time-Compression** acts as a powerful discount factor. The mathematical derivation shows that: $E'(0) < 0$. This implies that the energy decreases as we move from a V-shape ($\tau=0$) to a Y-shape ($\tau > 0$).
>
> Intuitively the Y-shape is cheaper not because it is shorter (it isn't),
> but because it allows the model to exploit the sublinear cost structure:
> it "waits" in a zero-cost trunk and then "sprints" the branches, which
> is energetically superior to the constant-speed V-shape trajectory.
>
> **Q 4-6**. To answer these questions, we have updated our method
> section, clarifying branching-related questions in more detail.

---

### Official Review · Reviewer_sui5 · 2025-11-09

**Soundness:** 3
**Presentation:** 2
**Contribution:** 3
**Rating:** 6
**Confidence:** 3

**Summary:**

Flow matching methods commonly use continuous-time flows with independent, straight-line trajectories -- V-shaped flows -- to transport simple distributions to data distributions, lacking mechanisms for trajectories to share transport. Authors argue that this uniform treatment overlooks the hierarchical and taxonomic structures of many real-world datasets. Inspired by branched transportation theory, they propose Y-shaped generative flows to enable adaptive, hierarchical transport: samples travel together initially, then branch to diverse targets. Empirically, Y-flows capture data hierarchies, reduce required integration steps, and improve distributional metrics over baseline flow models, offering a novel generative framework.

**Strengths:**

- Very clearly motivated, and very well-justified problem setup -- Standard flow matching and its alignment with optimal transport is in many ways too limited to capture real-world data and transitioning from independent, straight-line transport to branched, hierarchical movement inspired by branched optimal transport theory is significant. This represents a conceptual leap in generative modeling and addresses a limitation of current continuous-time flows that overlook hierarchical structure.

- The proposed velocity-powered objective is formally analyzed, with proofs showing its equivalence (up to constants) with flux-power costs under bounded density assumptions. The time-compression lemma provides elegant justification for improved computational efficiency.

- Instead of relying on computationally intractable classical branched transport formulations, the authors develop a neural ODE-based training objective and approximation procedure.

**Weaknesses:**

- The presentation could be improved; there are several typos across the paper. The technical constructions are mostly clear but Sec 4 could benefit from more explanations/intuitions.
- The theoretical guarantees rely on assumptions of bounded density, and the approach may favor near-instantaneous jumps or degenerate time-compression in less regular cases.
- Authors acknowledge that spatial-temporal regularization increases Jacobian computation and may be required for harder problems. There may remain edge-cases not fully addressed where non-smooth or singular data distributions could challenge this approach.

**Questions:**

Please see my review.

---

> ### Author Response · Authors · 2025-11-27
>
> We thank the reviewer for the thoughtful feedback and guidance in strengthening our paper. We have revised our paper to address the three highlighted weaknesses, which we will address below in the following order: Weakness 3, Weakness 2, and then Weakness 1.
>
> **W3. Authors acknowledge that spatial-temporal
> regularization increases Jacobian computation and may be required for
> harder problems..**:
>
> To address the computational and regularization concerns, we have explicitly integrated the Dirichlet energy regularization term $\lambda |\nabla_x v_\theta|^2$ into our core training objective (Eq. 7) and updated the theoretical analysis in Appendix D to reflect this dynamic formulation. While this involves Jacobian computations, we employ the Hutchinson trace estimator to ensure scalability, maintaining efficient training times (~1 hour for standard experiments). Additionally, we have empirically validated this regularized objective on higher-dimensional problems, including 250D and 685D single-cell data, with results reported in the new Tables 3 and 4. These experiments demonstrate that the method remains robust and computationally feasible even in high-dimensional settings without succumbing to singular edge cases.
>
> **W2. The theoretical guarantees rely on assumptions of
> bounded density, and the approach may favor near-instantaneous jumps or
> degenerate time-compression in less regular cases**:
>
> In our updated paper, we demonstrate that our formulation produces a branching
> structure without relying on bounded density assumptions. We have revised the theoretical framework to demonstrate that Y-shaped structures emerge from the energy competition itself, rather than relying on restrictive bounded density assumptions. We explicitly address the concern regarding degenerate time-compression in Section 4. While the sub-linear transport cost ($\alpha < 1$) does favor impulsive motion (Proposition 3, Time-Compression), this is counterbalanced by the cohesion regularization. We prove in Proposition 2 (The High Cost of Spatial Velocity Conflict) that instantaneous separation incurs a strictly positive cost proportional to the inverse of the support width. Consequently, Lemma 2 (Existence of Optimal Branching Time) establishes that the optimal trajectory has a Y-shaped structure.
>
> **W1. The presentation could be improved; there are
> several typos across the paper. The technical constructions are mostly
> clear but Sec 4 could benefit from more explanations/intuitions**:
>
> We have restructured Section 4 to improve clarity and intuition as suggested. We now begin by decomposing the static Modica-Mortola functional (Eq. 3) to explain the roles of the concave flux term and Dirichlet regularizer in isolation. We then systematically extend this to the dynamic setting (Eq. 5) and finally derive the tractable velocity-driven objective (Eq. 6). This narrative flow connects the theoretical foundations of branched transport directly to our practical Neural ODE implementation.

---

### Author Response · Authors · 2025-11-28
**Comments on the revised version**

We appreciate the reviewers' thoughtful feedback. We are excited that
the reviewers find our problem setup "very well-justified" and a
"conceptual leap in generative modeling" (sui5), and our theoretical
formulation "elegant and intuitive" (LMzY). We also value that the
reviewers recognize the novelty of applying branched flows
in a "stable and scalable way" (kJ38) and appreciate the
"cross-disciplinary contribution" regarding the shape of flows (ezAa).

Please consider the updated paper and appendices. The edits are
highlighted by the orange color in the revised version of the submission. The main edits are listed below:

-   **Refactored Theory (Section 4):** Prompted by the insightful
    remarks from kJ38 and sui5, we refactored Section 4. We
    established a theoretical link between the trade-off of
    Transport and Cohesion costs, culminating in the new **Lemma 2
    (Existence of Optimal Branching Time)**. This formally proves that
    our objective energetically favors Y-shaped trajectories
    ($\tau^* > 0$) over V-shaped ones.

-   **Practical Benefits & Efficiency:** Addressing the comments from
    LMzY and ezAa regarding practical application, we demonstrated the
    efficiency of our formulation by implementing an **early stopping
    criterion** in the Gaussian Mixtures experiment. Results show
    Y-Flows dynamically adapt, converging in $\sim3$ steps compared to
    the fixed 10 steps of Flow Matching.

-   **High-Dimensional Benchmarks (Appendix F):** We added **Tables 3
    and 4** to demonstrate robustness. These show that Y-Flows
    consistently outperform baselines (including Branched SBM and CNF)
    on biological datasets in dimensions up to 685D.

-   **Modica-Mortola Context (Appendix B):** We updated Appendix B to
    clarify the distinction between our scalable approach and the
    instabilities found in the classical Modica-Mortola framework (kJ38).

We believe that our formulation provides a valuable new perspective on
generative modeling, previously not established, by moving beyond
independent straight-line flows to reach the target in a fewer steps and capture the hierarchical nature of
real-world data. **This is particularly vital for naturally branched data,
such as biological differentiation, where our method improves upon
specifically designed baselines like BSBM**.

---

### Meta-Review · Area_Chair_mDB1 · 2026-01-08

**Summary:**

The paper proposes a generative modeling framework called Y-shaped Generative Flows. Most current continuous-time generative models (like Flow Matching or Diffusion) use "V-shaped" transport, where samples travel independently along straight lines from a prior (noise) to the data distribution. The authors argue this ignores the shared, hierarchical structure inherent in real-world data, such as biological cell differentiation.

I agree with reviewer LMzY regarding computation concern during training, as the method requires a simulation process for each of the training step, as explicitly discribed in #257, the reivewer is not likely to be satified with the response. also the provided evidence is not engough to support the claim that flowmatching produce "V-shaped" flows. There are multiple reference errors (#136, #258).

**Reviewer Concerns:**

The reviewer kJ38 was initially confused and then skeptical about the mathematical derivation. They argued that the proposed objective (Equation 7) appeared linear in density ($\rho$). In classical transport theory, if the cost is linear in density, there is no "economy of scale" incentive for paths to merge; branching or traveling separately would cost the same.

Reviewers questioned whether the model holds up in high-dimensional real-world settings and if the "fewer steps" claim was practically significant.

Reivewer LMzY have concerns regarding training efficient as it need to simulate the trajectory for each of the training step. Also LMzY questions the paper's central premise, that modern generative models like Flow Matching (FM) produce "V-shaped" flows.

**Reviewer Scores:**

kJ38 is likely to lower the socre, and LMzY is likely to maintain the original rating.

---

### Decision · Program_Chairs · 2026-01-26

Reject